# Autophagy inhibition prevents lymphatic malformation progression to lymphangiosarcoma by decreasing osteopontin and Stat3 signaling

Fuchun Yang[1], Shiva Kalantari[1], Banzhan Ruan[1], Shaogang Sun[1], Zhaoqun Bian[2] & Jun-Lin Guan [1]✉

Lymphatic malformation (LM) is a vascular anomaly originating from lymphatic endothelial cells (ECs). While it mostly remains a benign disease, a fraction of LM patients progresses to malignant lymphangiosarcoma (LAS). However, very little is known about underlying mechanisms regulating LM malignant transformation to LAS. Here, we investigate the role of autophagy in LAS development by generating EC-specific conditional knockout of an essential autophagy gene Rb1cc1/FIP200 in $Tsc1^{i\Delta EC}$ mouse model for human LAS. We find that $Fip200$ deletion blocked LM progression to LAS without affecting LM development. We further show that inhibiting autophagy by genetical ablation of FIP200, Atg5 or Atg7, significantly inhibited LAS tumor cell proliferation in vitro and tumorigenicity in vivo. Transcriptional profiling of autophagy-deficient tumor cells and additional mechanistic analysis determine that autophagy plays a role in regulating Osteopontin expression and its downstream Jak/Stat3 signaling in tumor cell proliferation and tumorigenicity. Lastly, we show that specifically disrupting FIP200 canonical autophagy function by knocking-in FIP200−4A mutant allele in $Tsc1^{i\Delta EC}$ mice blocked LM progression to LAS. These results demonstrate a role for autophagy in LAS development, suggesting new strategies for preventing and treating LAS.

Endothelial cells (ECs) are key components of both blood and lymphatic vessels, which are normally quiescent and tightly controlled by several signaling pathways, such as VEGF, Notch, and Eph signaling[1–6]. However, dysfunction of these pathways could result in abnormal proliferation of ECs, leading to a wide range of vascular anomalies, including vascular malformation and tumors[7–9]. Lymphatic malformation (LM) and lymphangiosarcoma (LAS) are vascular anomalies originating from lymphatic ECs. While LM mostly remains a benign disease, a fraction of LM patients can progress to LAS[10], a highly aggressive tumor currently without effective treatment with a reported 5-year survival rate of approximately 10%[11,12]. Although LM has been

recognized as a risk factor for the deadly LAS, the underlying mechanisms regulating the progression of LM to LAS is not well understood. Mechanistic target of rapamycin (mTOR) complex 1 (mTORC1) is a master regulator of cellular homeostasis[13,14], and dysfunction of mTORC1 signaling has been implicated in LM and LAS[15,16]. By generating an inducible EC-specific deletion of $Tsc1$, we recently developed a mouse model ($Tsc1^{i\Delta EC}$ mice) that exhibits LM and progression to LAS with characteristic features of human LAS[17]. This unique mouse model offers us opportunities to investigate the mechanisms of LM progression to LAS and develop effective prevention and treatments for LAS.

[1]Department of Cancer Biology, University of Cincinnati College of Medicine, Cincinnati, OH 45267, USA. [2]Department of Surgery, University of Cincinnati College of Medicine, Cincinnati, OH 45267, USA. ✉e-mail: guanjl@uc.edu

Autophagy is a conserved cellular process to sequester and deliver cytoplasmic materials to lysosomes for degradation and recycling[18,19]. Autophagy dysfunction is implicated in a variety of diseases including cancer[19–25], although nothing is known about the potential role and mechanisms of autophagy in the development of LM and progression to LAS. FIP200 (FAK-family Interacting Protein of 200 kDa) is a component of the ULK1/FIP200/Atg13/Atg101 complex, which is essential for the induction of autophagy[26]. By generating floxed FIP200 mice and taking a genetic approach using various mouse models, our research over the last decade have delineated multiple roles of autophagy in breast cancer and other biological and disease processes in vivo[27–35]. We showed that deletion of FIP200 decreased breast cancer development and metastasis using the MMTV-PyMT mouse model, providing the evidence for a pro-tumorigenesis role of autophagy in animals with an intact immune system[29]. Interestingly, however, FIP200 ablation in *BRCA1*-null mouse models of breast cancer did not inhibit tumor growth by itself, but only reduced tumor growth in vivo in combination with drugs that disrupt mitochondria by inhibiting mitochondrial biogenesis and sensitizing tumor cells to mitochondrial disrupting agent-induced cell death[36]. More recently, we showed that blocking autophagy by either FIP200 deletion or specifically block its canonical autophagy functions essentially alleviated the development of mammary tumors in the MMTV-Neu mouse model of breast cancer by perturbing the trafficking of oncogenic driver HER2 itself and promoting its release from tumor cells[37]. Moreover, although these and other studies by deleting different autophagy genes provided strong support for a pro-tumorigenic role of autophagy, some previous studies also showed a tumor suppressive role of autophagy in breast and other cancers[23–25]. Given that LM and its progression to LAS in *Tsc1*iΔEC mice are driven by mTORC1 hyperactivation[17], inhibition of autophagy downstream of activated mTORC1 could conceivably contribute to the development of LM and/or progression to LAS. On the other hand, recent studies revealed reverse positive regulation of mTORC1 by autophagy in neural stem cells and some other tumors[35,38,39], suggesting a potentially positive role for autophagy in LM development and progression to LAS by sustaining mTORC1 activation or through other mechanisms.

In this study, we demonstrated that autophagy blockade by deleting FIP200 or specifically disrupting its autophagy function, while not affecting LM development, abolished LM progression to LAS in vivo. Mechanistic studies using a new vascular tumor cell line derived from *Tsc1*iΔEC mice showed that blocking autophagy by depletion of FIP200, as well as knockout of two other autophagy genes Atg5 or Atg7, significantly inhibited vascular tumor cell proliferation in vitro and tumorigenicity in mice transplant assays. We further showed that autophagy blockade reduced expression of multiple genes in several signaling pathways, including Spp1 encoding Osteopontin (OPN), a multifunctional protein that regulates tumor cell proliferation, survival, and migration, and is implicated in promoting invasive and metastatic progression of many cancers[40–43]. Ectopic expression of OPN rescued decreased Jak/Stat3 signaling and reversed defective proliferation and tumorigenicity of FIP200-null vascular tumor cells, suggesting that autophagy-dependent OPN expression and its autocrine stimulation of Jak/Stat3 signaling contributes to the regulation of vascular tumor cells. Collectively, our studies reveal the role and mechanisms of autophagy in promoting LM progression to LAS, which will guide future design of effective prevention and novel therapies for this deadly disease.

## Results

### EC-specific FIP200 deletion prevents the progression of lymphatic malformation to LAS induced by mTORC1 hyperactivation

To study the role of autophagy in the development of LM and progression to LAS driven by *Tsc1* deletion and consequent mTORC1

hyper-activation in ECs, we crossed *Fip200*f/f mice[27] with *Tsc1*f/f;Scl-Cre mice[17] to generate *Tsc1*f/f;*Fip200*f/f;Scl-Cre mice for deletion of an essential autophagy gene *Fip200* in the previously described *Tsc1*iΔEC mouse model of human LAS[17]. Cohorts of *Tsc1*f/f;*Fip200*f/f;Scl-Cre and control littermates *Tsc1*f/f;*Fip200*+/+;Scl-Cre mice were treated with tamoxifen (TAM) at 8–10 weeks of age every other day for 3 times (2 mg each time) to induce activation of Cre recombinase to delete floxed *Tsc1* and *Fip200* genes in ECs (designated as 2cKO and *Tsc1*iΔEC mice, respectively). Analysis of lysates from the lung ECs of 2cKO and *Tsc1*iΔEC mice at 2 weeks following TAM treatment showed reduced levels of FIP200 in 2cKO mice, supporting at least partial deletion of the floxed Fip200 allele in ECs of these mice (Fig. 1a). Moreover, elevated levels of p62 were found in ECs from 2cKO mice compared to *Tsc1*iΔEC mice, as found in other cells after FIP200 deletion in previous studies[29,44]. Interestingly, we did not detect any increase in the phosphorylation of ULK1 at S757 by mTORC1 in ECs from either *Tsc1*iΔEC or 2cKO mice. No appreciable differences in LC3B levels were observed among ECs from these mice either. However, analysis of ECs from each cohort showed that the autophagy flux under starvation conditions in *Tsc1*iΔEC lung ECs was diminished upon ablation of FIP200 in lung ECs from 2cKO mice (Supplementary Fig. 1). These results suggest that autophagy is maintained despite Tsc1 deletion and mTORC1 activation in ECs, as we found in some other cells recently[35], and that ablation of FIP200 impairs autophagy flux in Tsc1 deleted ECs.

Similar to *Tsc1*iΔEC mice as observed previously[17], 2cKO mice developed LM as characterized by swelling in paws and tails at comparable frequency (Fig. 1b, c). However, in contrast to *Tsc1*iΔEC mice showing malignant LAS at 6–8 months after TAM[17], none of the 2cKO mice developed LAS even by 10 months after TAM (Fig. 1c, d). Histological examination of the edema lesions showed expanded vessels in both *Tsc1*iΔEC and 2cKO mice at 4 months after TAM (Fig. 1e), suggesting that FIP200 was not required for the LM phenotype in *Tsc1*iΔEC mice. At later time points (>6 months), we found solid tumor masses with dense cellularity typical of LAS with high proliferative index in *Tsc1*iΔEC mice, as previously observed[17], whereas the extended vessels with some differentiation typical of LM having low proliferative index were found in 2cKO mice (Fig. 1e, f left). We also detected increased apoptosis in 2cKO mice compared to *Tsc1*iΔEC mice (Fig. 1f right and 1g). Together, these results suggest the interesting possibility that *Fip200* ablation, while not affecting LM development, blocked LM progression to LAS in vivo.

### Blocking autophagy by inactivation of FIP200 or other autophagy genes decreases tumorigenicity of vascular tumor cells

To directly investigate the role and mechanisms of FIP200 and autophagy in the progression of LM to LAS, we employed shRNA mediated silencing of Fip200 in a spontaneously immortalized tumor cell line (designated as 562 cells) from tumors in *Tsc1*iΔEC mice. Of two different shRNA tested, shFip200-1 showed more efficient knockdown of FIP200 (Fig. 2a), and cells treated by it were pooled (designated as Fip200-KD cells) for further studies. Consistent with the results in vivo, Fip200-KD cells showed decreased cell proliferation (Fig. 2b), colony formation (Fig. 2c, d), and cell migration in wound healing assay (Fig. 2e, f). We then used xenotransplantation of nude mice to evaluate the effect of Fip200 inactivation on tumorigenicity of 562 cells in vivo. While parental 562 cells readily formed tumors when they were injected subcutaneously into nude mice, as previously observed[17], recipient mice injected with Fip200-KD cells did not develop any tumors or only generated very small tumors (Fig. 2g, h), suggesting that *Fip200* inactivation significantly compromised their tumorigenicity. Histological analysis of xenograft tumor sections showed markedly reduced tumor cell mass formed by Fip200-KD cells relative to control 562 cells (Fig. 2i). The reduced vascular tumor mass was verified by staining for CD31 in tumors formed by Fip200-KD cells, which also showed decreased proliferation and increased apoptosis as

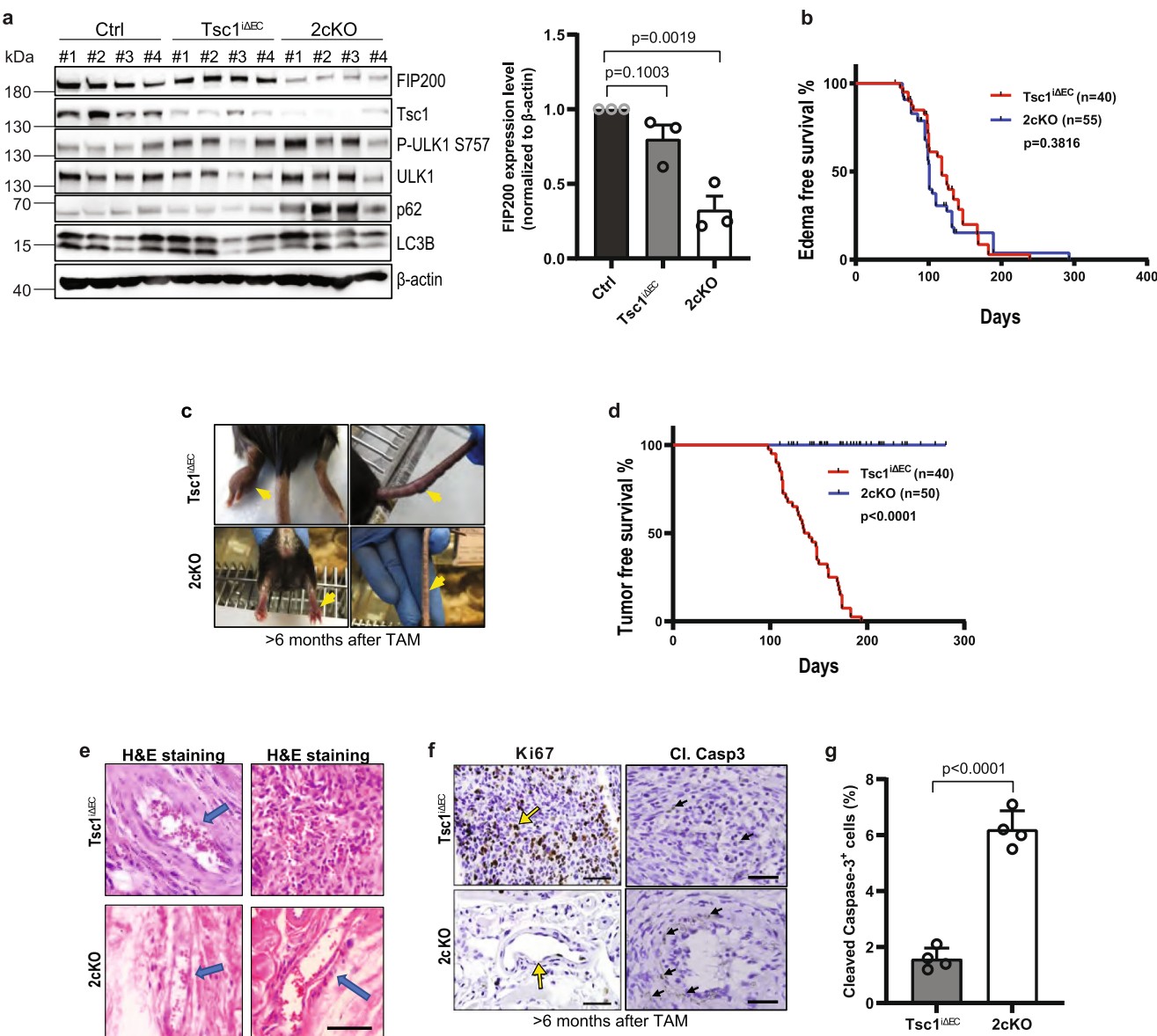

**Fig. 1 | *Fip200* ablation blocks LM progression to LAS. a** Lysates from lung ECs of wild type (Ctrl), *Tsc1*[iΔEC] mice and 2cKO mice (4 each) were analyzed by western blots for Tsc1, Fip200, p62, P-ULK1 S757, ULK1, LC3B and β-actin. Relative levels of Fip200/β-actin ratios (normalized to Ctrl cells) are shown as mean ± SD on the right. *n* = 3 independent experiments. **b** Kaplan-Meier analysis of edema development for *Tsc1*[iΔEC] (*n* = 40) and 2cKO (*n* = 55) mice. Log-rank (Mantel−Cox) test, *p* = 0.3816. **c** Representative images of tail and paw of *Tsc1*[iΔEC] mice with tumours and 2cKO mice with edema. Arrows mark tumors (*Tsc1*[iΔEC]) and lesions (2cKO). **d** Kaplan-Meier analysis of LAS for *Tsc1*[iΔEC] (*n* = 40) and 2cKO (*n* = 50) mice. Log-rank (Mantel−Cox) test, *p* < 0.0001. **e** Representative H&E staining of vascular lesions in the paws of *Tsc1*[iΔEC] and 2cKO mice at 4 and 6 months after TAM. Scale bar, 100 μm. Arrows mark lesions typical of LM. **f** Representative images of IHC for Ki67 and cleaved caspase 3 of vascular lesions in *Tsc1*[iΔEC] and 2cKO mice at 6 months after TAM. Scale bar, 100 μm. Arrow marks positive staining. **g** Quantification of percentage of cleaved caspase 3+ cells per field of view from *Tsc1*[iΔEC] and 2cKO mice at 6 months after TAM. *n* = 4 biological independent samples. Unpaired two-tailed t test was used in (**a**) (right panel) and (**g**). Log-rank (Mantel−Cox) test was used in (**b**) and (**d**). Source data are provided as a Source Data file.

measured by Ki67 and cleaved caspase 3 stainings, respectively (Fig. 2j–l). Interestingly, however, similar levels of phospho-S6RP were found in tumors formed by Fip200-KD and control 562 cells (Fig. 2j-third column). Western blotting analysis also showed comparable levels of S6K phosphorylation between Fip200-KD and control 562 cells (Fig. 2m). These results suggest that the decreased tumorigenicity after FIP200 knockdown was not due to decreased mTORC1 signaling and its consequent downstream events previously described[17].

To complement the studies above, we next used CRISPR-Cas9 to delete *Fip200* and other autophagy genes in 562 cells and examined effects on their proliferation and tumorigenicity. Both Fip200 KO-1 and Fip200 KO-2 cells (collectively called Fip200 KO cells) showed reduced proliferation and colony formation compared to control 562

cells in vitro (Fig. 3a–d). Moreover, *Fip200* KO completely abolished the ability of 562 cells to form tumors when injected into nude mice (Fig. 3e), likely due to the complete deletion of FIP200 in these cells versus a reduction in the amount of FIP200 in Fip200-KD cells. We also prepared 562 cells with knockout of two other essential autophagy genes, *Atg5* and *Atg7*, by CRISPR-Cas9 (Fig. 3f, g) for similar assays. We found that deletion of Atg5 or Atg7 also abolished tumorigenicity as measured by transplantation in nude mice (see Fig. 3e), as well as significantly reduced colony formation of 562 cells in vitro (Fig. 3h, i). To further extend these results, we employed CRISPR-Cas9 to knockout Fip200 in another spontaneously immortalized tumor cell line (designated as 5864 cells) from tumors in *Tsc1*[iΔEC] mice (Supplementary Fig. 2a). Similar to 562 cells, ablation of FIP200 decreased cell

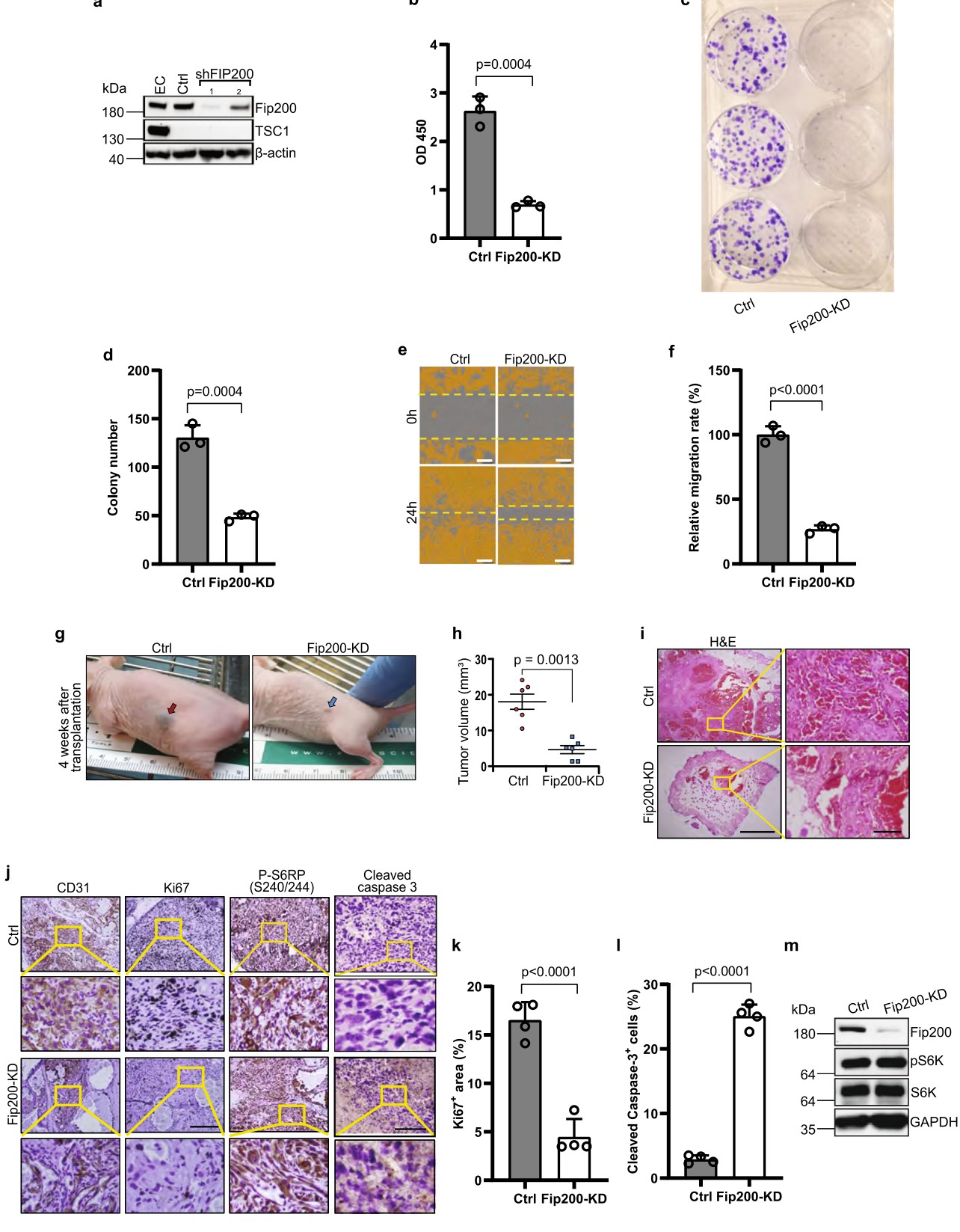

proliferation (Supplementary Fig. 2b), colony formation (Supplementary Fig. 2c), and cell migration in wound healing assay (Supplementary Fig. 2d) in 5864 cells. Deletion of FIP200 in 5864 cells also abolished tumorigenicity as measured by transplantation in nude mice (Supplementary Fig. 2e). These results provide further support that autophagy is required for tumorigenicity of vascular tumor cells.

**Autophagy blockade decreases osteopontin expression in vascular tumor cells**

To explore the mechanisms of autophagy in the regulation of vascular tumor cells, we examined changes in gene expression by transcriptional profiling of *Fip200* KO, *Atg5* KO, and *Atg7* KO cells. Three independent replicates of mRNA samples were prepared from each KO

**Fig. 2 | Fip200 knockdown decreases vascular tumour cell proliferation, migration and tumorigenicity. a** Lysates from mouse EC or 562 tumour cells treated with control or two separate Fip200 shRNA were examined by western blots with various antibodies as indicated. **b**–**f** 562 tumour cells and Fip200-KD cells were measured for cell proliferation (**b**), ncolony formation (**c**, **d**) and migration by wound healing assay (**e**, **f**). Data are presented as mean ± SD; *n* = 3 biologically independent cells. Scale bar in (**e**), 300 μm. **g**–**l** 562 tumour cells and Fip200-KD cells (10⁶ cells) were subcutaneously injected in recipient nude mice. Representative images of tumour formation at injection site at 4 weeks after injection are shown in (**g**), arrows mark tumour formation at the injection sites of Ctrl (red arrow) and Fip200-KD (blue arrow) mice, and Mean ± SEM of tumour volume are shown in (**h**). *n* = 6 biologically independent samples. Representative images of histological analysis by H & E staining, and IHC for CD31, Ki67, pS6RP and cleaved caspase 3 of tumour sections are shown in (**i**) and (**j**), respectively. The boxed area with yellow line is magnified. Scale bar, 500 μm (**i** left panel) and 50 μm (**i** right panel), 200 μm for CD31, Ki 67 and P-S6RP (S240/244) (**j**) and 50 μm for cleaved caspase 3 (**j**). Quantification of Ki67 positive cells per field of view (**k**) and percentage of cleaved caspase 3+ cells (**l**) in control and Fip200 KD transplanted tumours are shown as mean ± SD. *n* = 4 biologically independent samples. **m** Lysates from 562tumour cells and Fip200-KD cells were examined by immunoblots with various antibodies as indicated. Unpaired two-tailed *t* test was used in (**b**, **d**, **f**, **h**, **k**, and **l**). Source data are provided as a Source Data file.

cells and subjected to transcriptomic analysis (RNA-sequencing) in comparison to three samples from control 562 cells. We found that totals of 799, 1804, and 1652 genes were differentially expressed in *Fip200, Atg5*, and *Atg7* KO cells, respectively, compared to control 562 cells, with more down-regulated genes than up-regulated genes (Fig. 4a). KEGG pathway analysis of the differentially expressed genes identified significant changes in multiple pathways for each KO cells. Among these, PI3K-Akt signaling, ECM-receptor interaction and TNF signaling pathways were significantly changed in all three KO cells (Fig. 4b), indicating that these changes likely resulted from autophagy blockade (rather than the loss of possible autophagy-independent functions of specific genes). Furthermore, a set of 20, 9, and 10 genes were overlapped in PI3K-Akt signaling, ECM-receptor interaction, and TNF signaling, respectively, in all 3 KO cells (Fig. 4c and Supplementary Table 1). Among these genes, Spp1 gene encoding osteopontin (OPN) was represented in both PI3K/Akt signaling and ECM-receptor interaction pathways, suggesting its potentially prominent roles in mediating phenotypic effects of autophagy blockade in vascular tumor cells. Direct analysis of mRNAs by RT-qPCR confirmed reduced expression of Spp1, as well as several other genes in *Fip200* KO cells relative to control 562 cells (Fig. 4d and Supplementary Fig. 3). Reduced expression levels of OPN protein were also found in *Fip200* KO cells by Western blotting analysis (Supplementary Fig. 4), and ectopic expression of FIP200 rescued Opn levels in these cells (see Fig. 4e). The reduced Opn expression was also found in tumor sections of the recipient mice transplanted with Fip200-KD cells (Fig. 4f, g). Further, *Atg5* KO and *Atg7* KO cells also showed decreased expression of Spp1 (Fig. 4h and Supplementary Fig. 5) and Opn levels (Fig. 4i) compared to Ctrl cells. Together, these results demonstrate that autophagy blockade decreased OPN expression, which could contribute to the reduced tumorigenicity of vascular tumor cells.

## OPN mediates autophagy regulation of vascular tumor cells by affecting Stat3 signaling

Given the well-characterized functions of OPN in promoting cancer development and progression[40–43], the above results raised the interesting possibility that autophagy regulates vascular tumor cells through OPN signaling cascades. To evaluate such a hypothesis, we deleted Spp1 gene by CRISPR-Cas9 in 562 cells using two different sgRNA (OPN KO-1 and OPN KO-2 cells, collectively designated as OPN KO cells) (Fig. 5a). We found that OPN KO cells showed reduced colony formation (Fig. 5b, c) and tumor formation in xenograft transplantation (Fig. 5d), compared to control 562 cells. We next examined several signaling pathways[40,41,45], and found reduced Stat3 phosphorylation in OPN KO cells compared to 562 cells (Fig. 5e), which is consistent with previous studies in other cancers[46]. Moreover, autophagy blockade by depletion of FIP200, Atg5 or Atg7 also reduced phosphorylation of Stat3 (Figs. 4i and 5f). We also observed that phospho-Stat3 translocation into the nucleus (indicative of the activation of STAT3 pathway) was decreased upon autophagy blockade by depletion of FIP200, Atg5 or Atg7, and this decrease was reversed by the addition of Opn to Fip200 KO cells (Fig. 5g). To further extend these results, we employed CRISPR-Cas9 to knockout Spp1 in 5864 cells (Supplementary Fig. 6a).

Similar to 562 cells, ablation of OPN decreased cell proliferation (Supplementary Fig. 6b), colony formation (Supplementary Fig. 6c, d), tumorigenicity as measured by transplantation in nude mice (Supplementary Fig. 6e), and Stat3 phosphorylation (Supplementary Fig. 6f) in 5864 cells. Together, these results suggest that decreased Jak/Stat3 signaling downstream of OPN may contribute to the decreased tumorigenicity of vascular tumor cells.

To further assess the role of autophagy-dependent expression of OPN and its stimulation of Jak/Stat3 signaling in vascular tumor cells, we infected *Fip200* KO cells with recombinant lentiviruses encoding OPN and examined the effects of ectopic OPN expression in these autophagy-deficient cells. Consistent with a role for OPN in mediating the effects upon Fip200 deletion, we found that phosphorylation of Stat3 was also significantly decreased in Fip200 KO cells (Fig. 6a). Importantly, re-expression of OPN rescued defective Stat3 phosphorylation (Fig. 6a), tumor cell proliferation as measured by colony formation assay (Fig. 6b, c), as well as tumor cell migration in wound healing assays (Fig. 6d, e). Consistent with these results in vitro, xenograft transplantation assays in recipient nude mice showed that OPN re-expression also restored tumorigenicity of Fip200 KO cells (Fig. 6f, g). Immunohistochemical analysis of tumor sections from the recipient mice also showed rescue of phosphorylation of Stat3 and tumor cell proliferation as measured by Ki67 staining by ectopic expression of OPN in Fip200 KO cells (Fig. 6h, i). Together, these results suggest a role of OPN stimulation of Jak/Stat3 signaling in mediating the regulation of vascular tumor cell proliferation and tumorigenicity by autophagy.

## Specific ablation of FIP200 canonical autophagy functions blocks LM progression to LAS

Our above analysis using knockout of several autophagy genes including Fip200 as well as Atg5 and Atg7 in 562 cells provided strong support for the regulation of vascular tumor cell growth by autophagy per se, rather than the loss of other potential autophagy-independent functions of these genes even though increasing evidence suggests that most, if not all, autophagy genes have functions independent of their roles in canonical autophagy[47–49]. Nevertheless, these studies do not address the role of Atg5 or Atg7 directly in LM progression to LAS in vivo. We have previously taken a rigorous genetic approach to generate *Fip200-4A* mutant knock-in allele that blocks its autophagy function specifically to reveal both autophagy and non-autophagy functions of FIP200 in breast cancer and other diseases in vivo[35,48,50]. To examine the specific role of FIP200-mediated autophagy in LM progression to LAS in vivo, we crossed *Fip200 + /KI* mice containing *Fip200-4A* allele to *Tsc1*ᶠ/ᶠ;*Fip200*ᶠ/ᶠ;Scl-Cre mice to generate *Tsc1*ᶠ/ᶠ;*Fip200*ᶠ/ᴷᴵ;Scl-Cre mice. Treatment of these mice with TAM induced deletion of floxed *Tsc1* and *Fip200* alleles, leading to the expression of only the FIP200-4A mutant lacking autophagy function specifically in Tsc1-deficient ECs (designated as 2cKI mice). Analyses of this mouse model showed that specifically blocking FIP200 autophagy function, while not affecting the development of LM (Fig. 7a), abrogated their progression to LAS (Fig. 7b) in a manner similar to FIP200 ablation in *Tsc1*ᶦᐞᴱᶜ mice (see Fig. 1b, d).

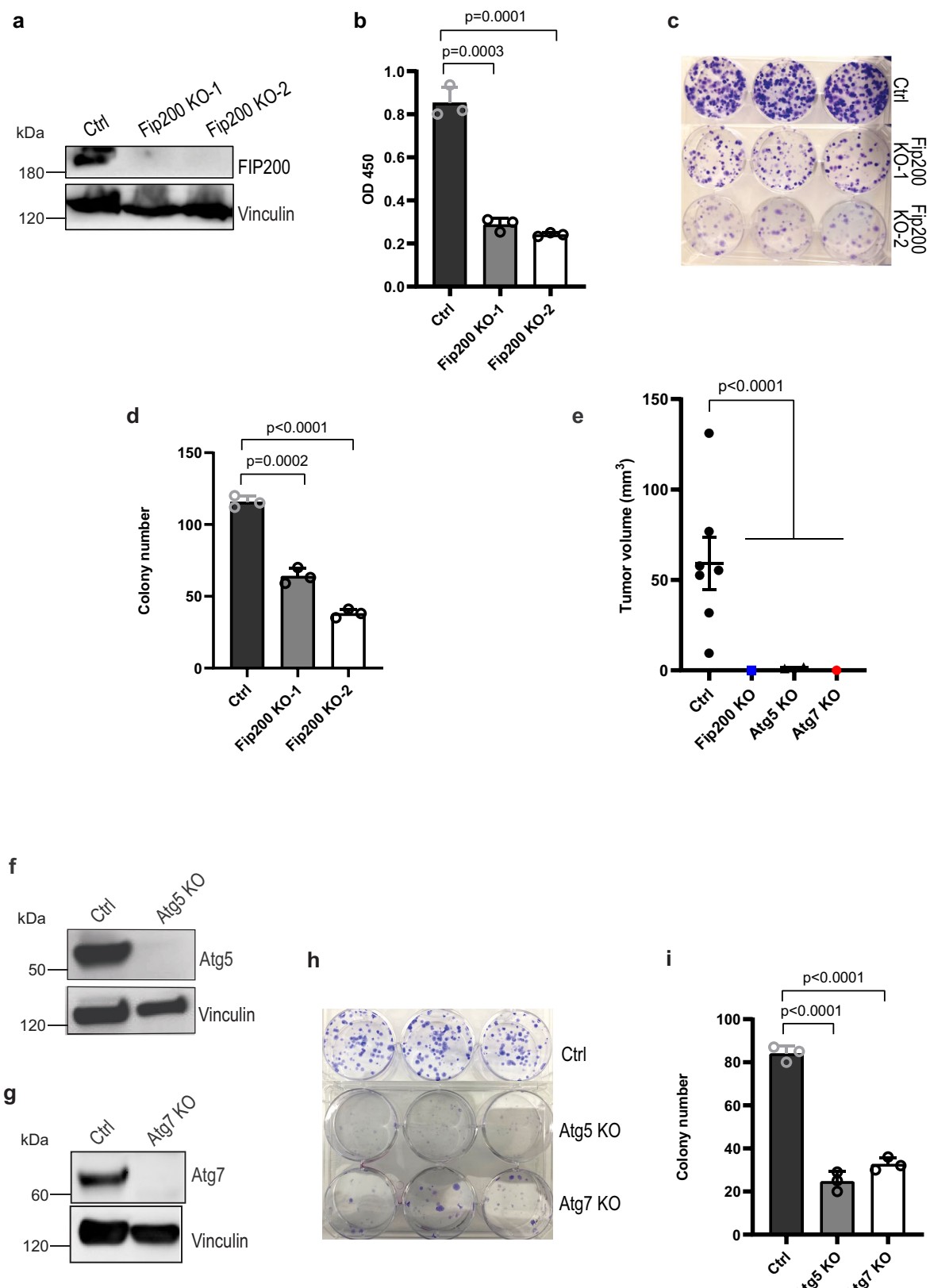

These results are supported by histological analysis of vascular lesions in 2cKI mice (Fig. 7c). Lastly, similarly reduced levels of phosphorylated Stat3, Ki67 and OPN staining were found in 2cKI mice and 2cKO mice relative to $Tsc1^{i\Delta EC}$ mice (Fig. 7d–g). These results provide further support that autophagy is required for the progression of LM to LAS in $Tsc1^{i\Delta EC}$ mice, suggesting that targeting canonical autophagy could be effective in preventing and treating the deadly LAS.

## Discussion

Autophagy inhibition is increasingly recognized as a potential treatment for various cancers, although there is still debate on the

**Fig. 3 | Autophagy gene knockout decreases vascular tumour cell colony formation and tumorigenicity. a** Lysates from 562 tumour cells (Ctrl) and two clones of these cells with CRISPR-Cas9 mediated *Fip200* knockout (KO) were examined by western blots for FIP200 and Vinculin. **b** Ctrl and Fip200 KO tumour cells were measured for cell proliferation by CCK-8 assay shown as mean ± SD. $n = 4$ biologically independent cells. **c, d** Representative images of colony formation assay of Ctrl and Fip200 KO tumour cells are shown in **c**, and mean ± SD of the colony number are shown in **d**. $n = 3$ biologically independent cells. **e** Ctrl, Fip200 KO, Atg5 KO and Atg7 KO tumour cells ($1 \times 10^6$ cells) were subcutaneously injected in

recipient nude mice. Mean ± SEM of tumour volume at 3 months after injection are shown. $n = 4$ mice for each group. **f, g** Lysates from Ctrl, Atg5 KO (**f**) and Atg7 KO (**g**) tumour cells were examined by western blots using various antibodies. **h, i** Representative images of colony formation assay of Ctrl, Atg5 KO and Atg7 KO tumour cells are shown in **h**, and mean ± SD of the colony numbers are shown in (**i**). $n = 3$ biologically independent cells. Unpaired two-tailed $t$ test was used in (**b, d**, and **i**), one-way ANOVA with Dunnett's multiple comparisons test in (**e**). Source data are provided as a Source Data file.

possibility of a suppressive role of autophagy for some cancers and/or in different context[23–25]. It is generally thought that autophagy may suppress initial tumor development by helping to maintain genomic stability, while it has a positive role for tumor growth and progression by supplying various metabolites and nutrients to the increased demand of tumor cells. In this study, we showed that both deletion of FIP200 and specifically disrupting its autophagy function abrogated the development of malignant LAS from the benign LM lesions, suggesting that inhibiting autophagy could potentially prevent tumorigenesis (rather than induce it) for this deadly disease. Our data also suggested that autophagy-dependent expression of OPN and its autocrine stimulation of Jak/Stat3 signaling contribute to the promotion of vascular tumor cell proliferation and tumorigenicity by autophagy. Given that LM is a risk factor for LAS and the lack of effective treatment for the deadly LAS, potential new prophylactic strategies based on our findings to targeting autophagy and/or its downstream OPN expression will have important clinical implications for LM patients.

Previous studies using mouse models and in human patients both indicated a critical role for hyper-activation of mTORC1 signaling in vascular malformation and tumors including LM and LAS[17]. Although mTORC1 activation is known to suppress autophagy through phosphorylation of ULK1 at Ser757[51,52], recent studies also suggested that selective autophagy such as lipophagy could play a role for sustaining mTORC1 activation and fulfilling increased bioenergetic demands in Tsc-deficient neural stem cells, as well as some tumor cells under energy stressed conditions[35,38]. Nevertheless, we did not find significant changes in mTORC1 activation upon autophagy blockade by FIP200 deletion, suggesting that it is unlikely that autophagy inhibition abrogated LM progression to LAS by directly affecting mTORC1 as a driver in vascular tumor cells.

OPN is a multifunctional protein that regulates tumor cell proliferation, survival, and migration, and is implicated in promoting invasive and metastatic progression of many cancers[40–43]. Our studies using both mouse models in vivo and vascular tumor cells in vitro suggest a role for OPN to mediate regulation of vascular tumor cells by autophagy. Several lines of evidence support that autophagy-dependent expression of OPN and its autocrine stimulation of Jak/Stat3 signaling contribute to the promotion of vascular tumor cell proliferation and tumorigenicity by autophagy. Ablation of *Fip200* and other autophagy genes reduced expression of OPN and its downstream Jak/Stat3 signaling pathway. Conversely, re-expression of FIP200 in Fip200 KO cells restored OPN levels and Stat3 signaling as measured by both its phosphorylation and translocation to the nucleus. Moreover, ectopic expression of OPN rescued decreased Jak/Stat3 signaling and reversed defective proliferation and tumorigenicity of FIP200-null vascular tumor cells. Besides Stat3 signaling, we also observed decreased mTORC1 signaling in OPN KO tumor cells (Supplementary Fig. 7), suggesting that OPN may regulate vascular tumor cells through other signaling pathways like mTORC1. Although mTORC1 signaling was not reduced after depletion of FIP200 (see Figs. 2j and 2m), additional studies will be needed to clarify the potential contribution of mTORC1 signaling in the decreased tumorigenesis of vascular tumor cells upon autophagy blockade and reduced OPN expression. In addition to acting as an extracellular ligand

through binding to cell surface receptors, OPN has been suggested to possess intracellular roles to regulate various cell functions by its different isoforms[42,43,53,54]. It will be interesting to carry out systematic analysis of different isoforms of OPN by both ectopic expression (acting both intracellularly and extracellularly) and as extracellular supplementation (acting only extracellularly) to determine which isoforms and/or cellular modes play preferential roles in mediating autophagy regulation of vascular tumor cell functions in future studies. It will also be interesting to determine whether autophagy regulation of OPN is important in the promotion of tumor growth and/or metastasis by autophagy in other cancers.

In addition to OPN, our results showed down-regulations of other genes in several signaling pathways including PI3K-Akt signaling, ECM-receptor interaction and TNF signaling in vascular tumor cells after autophagy blockade. PI3K-Akt signaling has been well studied in vascular and other tumors, including our previous studies establishing a *Tsc1*$^{iΔEC}$ mice model[17,55,56]. ECM-receptor interactions are also known to affect development and progression of different cancers[57,58], although their potential roles in vascular tumor cells are not well understood. Integrins are the major cell surface receptors for ECM[59], and several integrins including αvβ3 and αvβ5 have been shown to function as receptors for OPN[60]. Interestingly, *Itgb5* encoding β5 subunit of αvβ5 integrin is among the 9 genes in ECM-receptor interaction that were down-regulated in all three *Fip200, Atg5, Atg7* KO cells (see Fig. 4c). It will be interesting to examine the possibility that decreased expression of integrin β5 and/or other genes upon autophagy blockade could contribute to the reduced proliferation and tumorigenicity in coordination with the decreased OPN in vascular tumor cells in future studies.

Autophagy has been recognized to play both pro-tumorigenesis and tumor suppressive roles in the development and progression of various cancers[23–25], likely due to its impact on many cellular functions in different contexts. Our results suggest that autophagy could play a differential role even within the same lymphatic ECs with hyper-activation of mTORC1 signaling, i.e. while disposable for the development of LM phenotype, but required for LM progression to LAS. Autophagy blockade also inhibited growth of tumor cells from LAS in vitro and in xenograft transplant recipients, although it is still not clear whether autophagy may be necessary to promote LAS growth and/or metastasis in *Tsc1*$^{iΔEC}$ mice, as no spontaneous LAS were detected in 2cKO or 2cKI mice. New mouse models to allow ablation of *Fip200* or other autophagy genes after the deletion of *Tsc1* to induce LAS formation (i.e. unlike simultaneous deletion in the 2cKO mice) will be necessary for these studies. In another recent study, we found that HDAC inhibition triggered vascular tumour autophagic cell death in vitro and decreased tumor growth in xenograft models by increasing autophagy activity in these cells[61]. Together with the current study, these results suggest that either blocking or inducing aberrantly high levels of autophagy could potentially inhibit vascular tumor growth, further highlighting the critical role of autophagy in vascular tumor cells.

We observed previously that tumor cells isolated from LAS in *Tsc1*$^{iΔEC}$ mice can form vascular tumors by transplantation to recipient nude mice[17]. However, ECs isolated from lungs of the same mice could not form tumors in the same assay, suggesting that *Tsc1* deletion alone is not sufficient to induce LAS, although it is required for the

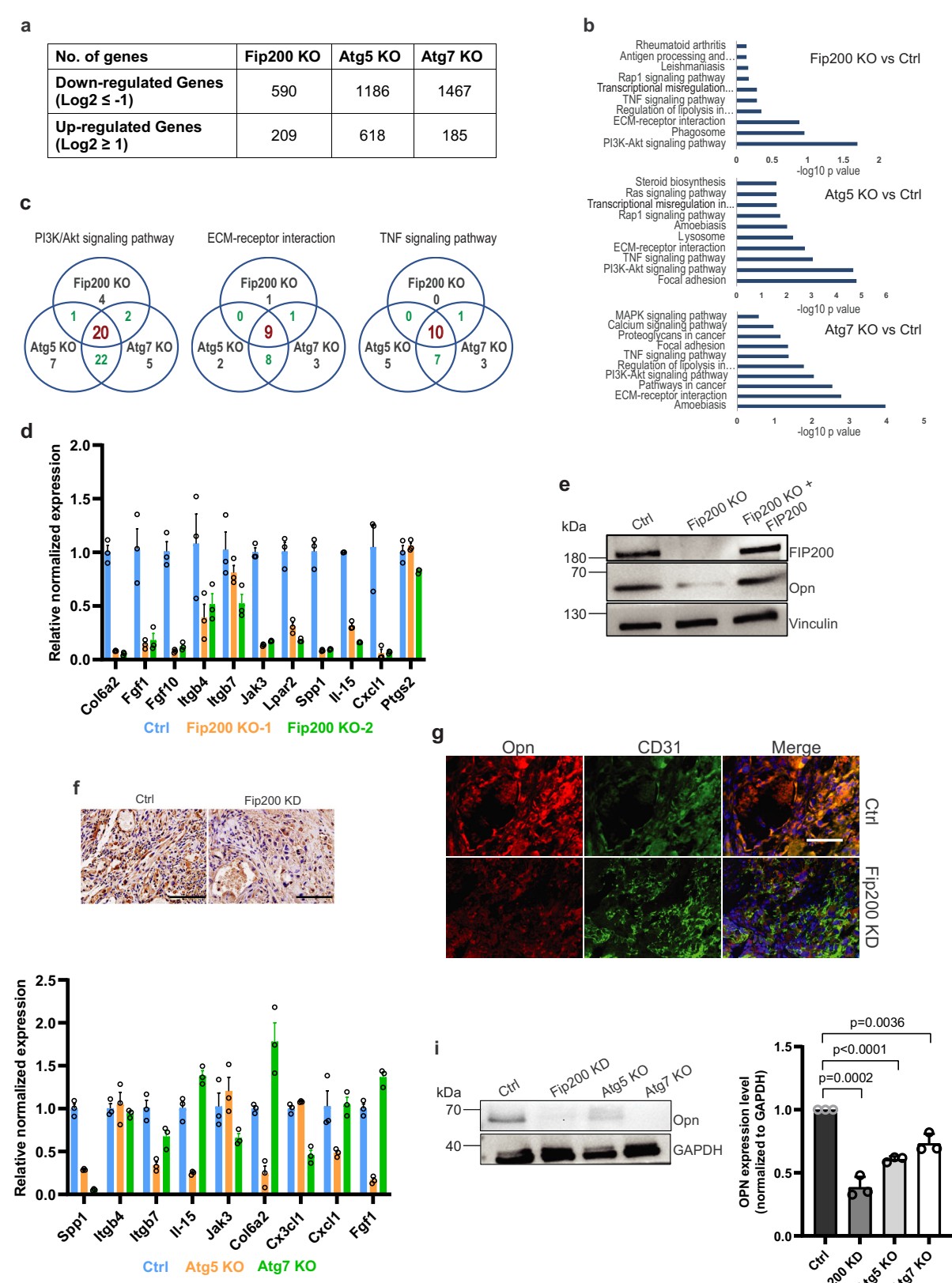

process. It is possible that abnormalities in other signaling pathways and cellular processes in ECs may be necessary for LM development and/or progression to LAS. Interestingly, previous clinical studies reported that malignant progression to angiosarcoma from vascular malformation in some human patients were accompanied by additional mutations[10], even though it is not clear about the specific

contribution of these mutations to the development or maintenance of angiosarcoma. These observations are consistent with our results that autophagy blockade reduced OPN signaling through Jak/Stat3 pathway contributing to the reduced vascular tumor cell growth, without impacting on mTORC1 signaling per se. Likewise, the altered PI3K-Akt signaling, ECM-receptor interaction and/or TNF signaling

**Fig. 4 | Transcriptome profiling identifies decreased OPN expression in vascular tumour cells after autophagy blockade. a** Number of down-regulated and up-regulated genes in Fip200 KO, Atg5 KO and Atg7 KO tumour cells compared to Ctrl tumour cells. **b** KEGG pathway analysis plots showing changes in different pathways in Fip200 KO, Atg5 KO and Atg7 KO tumour cells versus Ctrl tumour cells. **c** Number of down-regulated genes in the PI3K-Akt signaling, ECM-receptor interaction and TNF signaling pathways in Fip200 KO, Atg5 KO and Atg7 KO tumour cells are shown with the common genes for all 3 KO cells highlighted in red and those overlapping in 2 KO cells highlighted by green. **d** A select group of down-regulated genes including Spp1 in Fip200 KO cells were validated using RT−qPCR. Mean ± SEM of relative levels (normalized to Ctrl cells) is shown. *n* = 3 technical replicates. **e** Lysates from Ctrl and Fip200 KO tumour cells with or without re-expression of FIP200 were examined by western blots for FIP200, OPN and Vinculin. **f** Representative images of IHC for OPN of tumour sections in mice transplanted with 562 tumour cells and Fip200-KD cells at 3 months after injection. Scale bars, 100 μm. **g** Representative images of immunofluorescent staining for OPN and CD31 of tumour sections in mice transplanted with 562 tumour cells and Fip200-KD cells at 3 months after injection. Scale bars, 50 μm. **h** A select group of down-regulated genes including Spp1 in Atg5 KO and Atg7 KO cells were validated using RT−qPCR. Mean ± SEM of relative levels (normalized to Ctrl cells) is shown. *n* = 3 technical replicates. **i** Lysates from Ctrl, Fip200-KD, Atg5 KO and Atg7 KO tumour cells were examined by western blots for OPN and GAPDH. Relative levels of Opn (normalized to Ctrl cells) are shown as mean ± SD on the right. *n* = 3 independent experiments. Bonferroni test was used in (**b**). Unpaired two-tailed t test was used in **i** (right panel). Source data are provided as a Source Data file.

based on transcriptional profiling may suggest their potential synergistic role with mTORC1 hyper-activation in LM progression to LAS in autophagy-dependent manner. These could explain the differential functions of autophagy in LAS but not LM found in this study, although future studies will be needed to clarify the underlying mechanisms, which will be important to guide potential clinical applications and reap full benefit of targeting autophagy in the prevention and treatment of LAS.

## Methods

### Mice

Mice were housed and handled according to local, state, and federal regulations, and all experimental procedures were carried out as per the guidelines of the Institutional Animal Care and Use Committee (IACUC) at the University of Cincinnati. Animal husbandry including housing location and procedure located at Laboratory Animal Medical Services (LAMS) and was supported by LAMS staff. The housing conditions for the mice: the light cycle is 14:10 (light /dark cycle) or 6am ON/8 pm Off, temperature setpoint is 74 °F with a +/− 4 °F, and humidity range is between 30−70%. All mice were finally euthanized with medical carbon dioxide and cervical dislocation at the end point time. *Tsc1*^f/f^;Scl-Cre mice[17] with *Fip200*^f/f^ mice[27] to obtain *Tsc1*^f/+^;*Fip200*^f/+^;Scl-Cre and *Tsc1*^f/+^;*Fip200*^f/+^ mice, which were then inter-crossed to generate *Tsc1*^f/f^;*Fip200*^f/f^;Scl-Cre mice and control littermates *Tsc1*^f/f^;*Fip200*^+/+^;Scl-Cre and *Tsc1*^f/f^;*Fip200*^f/+^;Scl-Cre mice. *Tsc1*^f/f^;*Fip200*^f/f^;Scl-Cre mice with *Fip200*^KI/+^ mice[48] to obtain *Tsc1*^f/+^;*Fip200*^f/KI^;Scl-Cre and *Tsc1*^f/+^;*Fip200*^f/+^ mice, which were then inter-crossed to generate *Tsc1*^f/f^;*Fip200*^f/KI^;Scl-Cre. These mice have 98% C57BL/6 backgrounds. The above obtained mice including both male and female were intraperitoneally injected with tamoxifen (TAM) at 8−10 weeks of age every other day for 3 times (2 mg each time) to induce activation of Cre recombinase to delete floxed *Tsc1* and *Fip200* genes in ECs (designated as 2cKO and *Tsc1*^iΔEC^ mice, respectively), as well as one allele Fip200 deletion and Fip200-4A mutant knock-in mice (designated as 2cKI mice). Xenograft transplantation was performed by subcutaneously injected with tumor cells in 6−8 weeks old female athymic nude mice. Tumor size was measured at the endpoints and calculated according to the formula volume = (length × width$^2$) / 2. Tumor volumes were analyzed using GraphPad Prism 9 software. The maximum footpad or subcutaneous tumour size of 5 mm in any direction and the maximum transplanted tumour size with 20 mm longest diameter are permitted by IACUC at University of Cincinnati, and the maximum tumour size was not exceeded in all experimental data.

### Generation and culture of *Tsc1*^iΔEC^ vascular tumor cells and primary lung endothelial cells

Vascular tumor cells were isolated from the cutaneous tumors in Tamoxifen induced *Tsc1*^f/f^;Scl-Cre mouse, and cultured in 6-well plate for several months to establish spontaneously immortalized *Tsc1*^iΔEC^ tumor cells. Primary lung ECs were isolated from lungs of Tamoxifen-induced *Tsc1*^f/f^;Scl-Cre and *Tsc1*^f/f^;*Fip200*^f/f^;Scl-Cre mice, and cultured in 6-well plate for 2 weeks to obtain enough cells, using rat anti−murine CD31 (MEC 13.3, BD) coated magnetic beads (M-450; sheep anti−rat IgG Dynabeads, Invitrogen) as described previously[17,62]. All cells were cultured in EC culture medium and grew in 5% CO2 at 37 °C. For autophagy flux assays, lung EC cells were cultured in EBSS starvation without serum for 4 h with or without addition of 200 nM Bafilomycin A1 (Bal-A1) (Cayman #11038) for the last 2 h. Quantification of autophagy flux was calculated as LC3-II/ Vinculin with Bal-A1 subtracting LC3-II/Vinculin without Baf-A1 under EBSS starvation. The autophagy flux of control was set as 1, data was plotted as mean ± SD of Tsc1cKO and 2cKO normalized to control from three independent experiments. All cells were tested for negative mycoplasma contamination.

### Plasmids and generation of stable cell lines

shRNAs targeting mouse Fip200 5′-GCTGAATTTCAGTGCTTAGAA-3′ (TRCN0000084986) and 5′-CCAACTTTAACACAGTCTTAA-3′ (TRCN0 000084987) were purchased from Sigma-Aldrich. A scrambled shRNA as control was purchased from Addgene (plasmid #1864). sgRNAs targeting mouse Fip200 (5′-TTCTCTAGAAATAACACTAA-3′; 5′-CTCCA TTGACCACCAGAACC-3′), Atg5 (5′-CCCTATAGACCACGACGGAG-3′), Atg7 (5′-GAACGAGTACCGCCTGGACG-3′) and Opn (5′-GCAAATCACT GCCAATCTCA-3′; 5′-GCAGAATCTCCTTGCGCCAC-3′) were respectively cloned into LentiCRISPR v2 vector (Addgene #52961) according to the protocol previously reported[63]. sgRNA targeting EGFP (Addgene #86153) was used as control. Spp1 overexpression clone was constructed by RT-PCR from HUVEC RNA and then insertion into pLV-EF1a-IRES-Neo vector (Addgene #85139).

All lentiviral productions were generated by transfecting into HEK293T cells with psPax2, pMD2.G and lentiviral plasmids, as described previously[64]. *Tsc1*^iΔEC^ cells were then transduced by different lentivirus containing specific sgRNA followed with puromycin selection for one week. The efficiencies of Fip200 knockdown, Fip200, Atg5 and Atg7 knockout or re-overexpression of osteopontin in tumor cells were confirmed by western blot.

### Cell proliferation and colony formation assay

Cell proliferation was assessed by Cell Counting Kit-8 (CCK-8, Dojindo Molecular Technologies) according to their manual protocols. $2 × 10^4$ cells were seeded in 96-well plate with triplicates for each experimental group, cells were analyzed after 72 h.

For colony formation assay, 1000 cells were seeded in 6-well plates with triplicates, then stained with 2% crystal violet solution after 7 days.

### Cell migration assay

A total of 40,000 cells were seeded in 96-well ImageLock tissue culture plate (Essen BioScience 4379). Next day, cells in each well were scratched using WoundMaker (Essen BioScience) and washed twice with culture medium. Fresh culture medium were then added and the plates

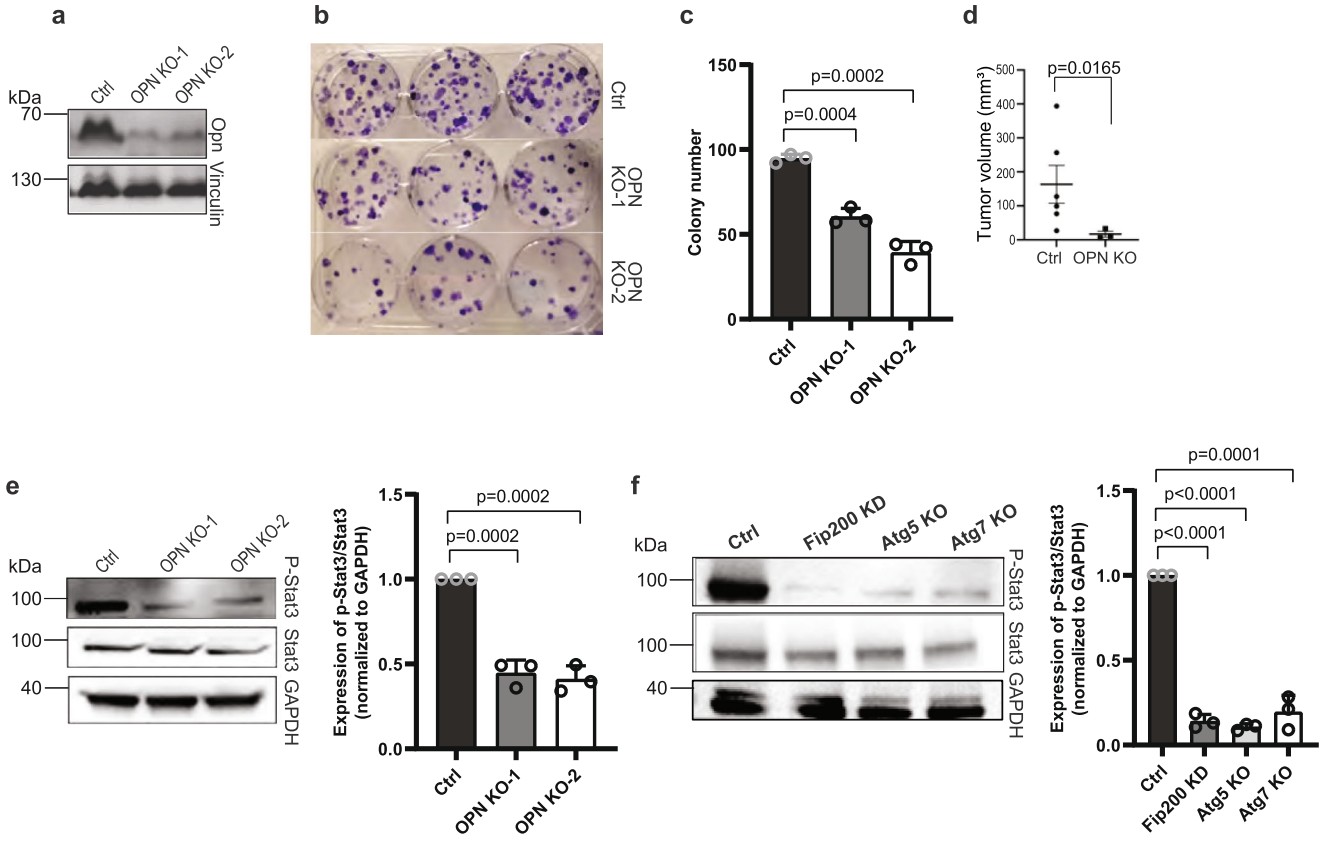

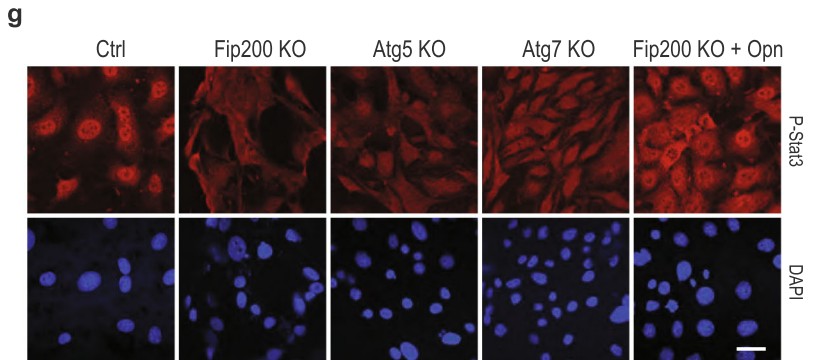

**Fig. 5 | OPN knockout inhibits vascular tumour cell colony formation, tumorigenicity and Stat3 phosphorylation. a** Lysates from Ctrl and two clones of OPN KO tumour cells were examined by western blots for OPN and Vinculin. **b**, **c** Representative images of colony formation assay of Ctrl and OPN KO tumour cells are shown in **b**, and mean ± SD of the colony number are shown in **c**. $n = 3$ biologically independent cells. **d** Ctrl and OPN KO tumour cells were subcutaneously injected in recipient nude mice. Mean ± SEM of tumour volume at 10 weeks after injection is shown. $n = 3$ mice for each group. **e** Lysates from Ctrl and OPN KO tumour cells were examined by western blots for P-Stat3, Stat3 and GAPDH. Relative levels of pSTAT3/STAT3 (normalized to Ctrl cells) are shown as mean ± SD on the right. $n = 3$ independent experiments. **f** Lysates from Ctrl, Fip200-KD, Atg5 KO and Atg7 KO tumour cells were examined by western blots for P-Stat3, Stat3 and GAPDH. Relative levels of pSTAT3/STAT3 (normalized to Ctrl cells) are shown as mean ± SD on the right ($n = 3$ independent experiments). **g** Representative images of immunofluorescent staining for p-STAT3 and DAPI of Ctrl, Fip200 KO, Atg5 KO, Atg7 KO cells and Fip200 KO tumour cells with OPN re-expression (Fip200 KO + OPN), as indicated. Scale bar, 10 μm. Unpaired two-tailed t test was used in **c**, **d**, **e** (right panel), and **f** (right panel). Source data are provided as a Source Data file.

were placed in IncuCyte ZOOM for image scanning every 2 h for 24 h using 20 X objective.

## RT-qPCR

Total RNAs were isolated from cells with GeneJET RNA purification Kit (Thermo Scientific #K0731) according to user manual. Reverse transcription cDNAs were synthesized with iScript cDNA Synthesis Kit (Bio-Rad #1708891). Real-time PCR was performed with iQ SYBR Green Supermix Kit (Bio-Rad #170-8880). Expression values were normalized to β-actin. The primers were obtained from PrimerBank (https://pga.mgh.harvard.edu/primerbank/) unless specific references were cited. The specificity of all primers was validated with their dissociation curves. The sequence (5′–3′) of each pair of primers are listed as follows.

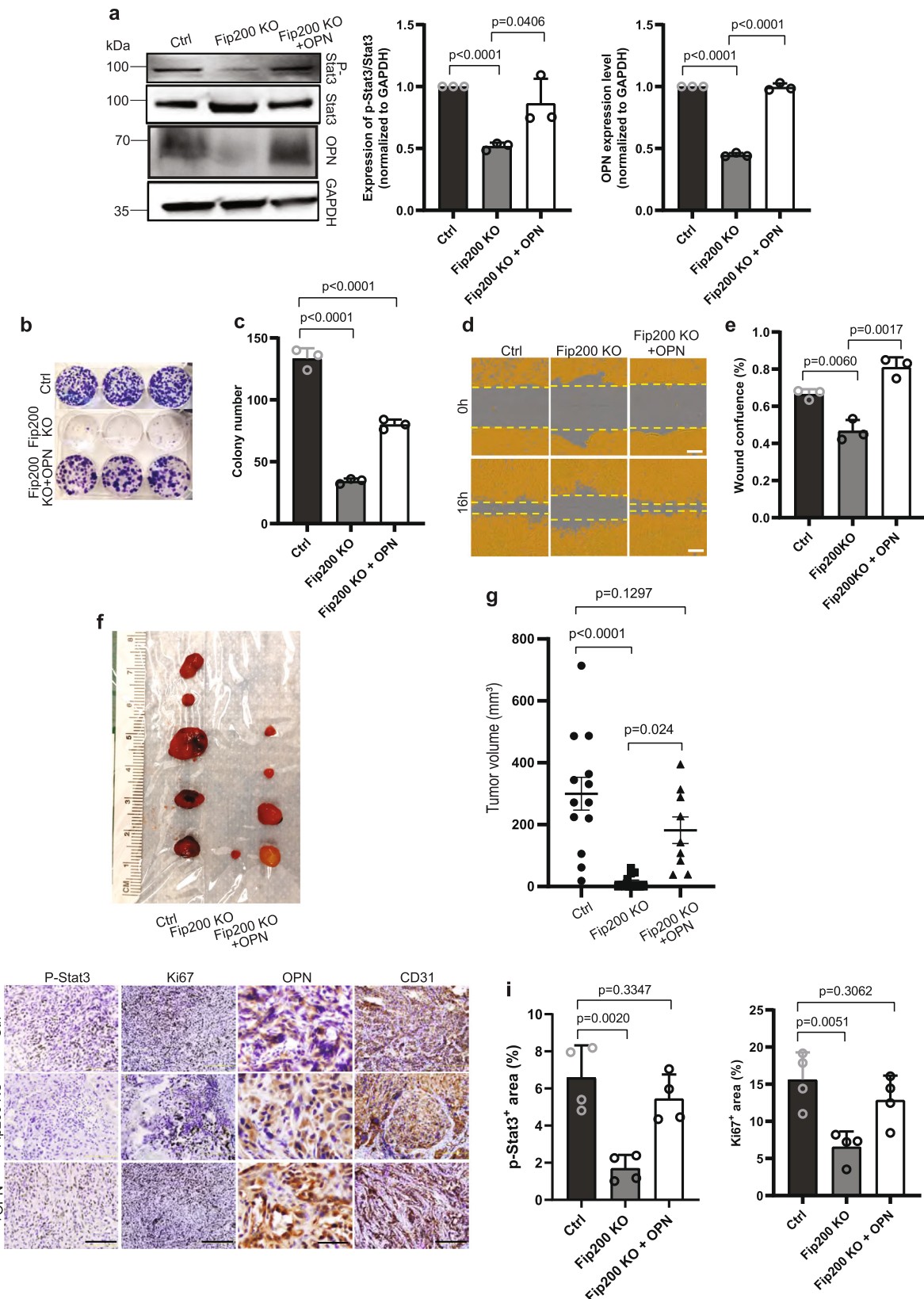

Spp1-Forward: ATCTCACCATTCGGATGAGTCT;
Spp1-Reverse: TGTAGGGACGATTGGAGTGAAA.
Col6a2-Forward: AAGGCCCCATTGGATTCCC;
Col6a2- Reverse: CTCCCTTCCGACCATCCGAT.
Fgf10-Forward: TTTGGTGTCTTCGTTCCCTGT;
Fgf10-Reverse: TAGCTCCGCACATGCCTTC.

Cxcl1-Forward: CTGGGATTCACCTCAAGAACATC;
Cxcl1-Reverse: CAGGGTCAAGGCAAGCCTC.
Itgb4-Forward: GCAGACGAAGTTCCGACAG;
Itgb4-Reverse: GGCCACCTTCAGTTCATGGA.
Itgb7-Forward: ACCTGAGCTACTCAATGAAGGA;
Itgb7-Reverse: CACCGTTTTGTCCACGAAGG.

**Fig. 6 | Ectopic expression of OPN rescues deficient colony formation, migration, tumorigenicity and Stat3 phosphorylation of Fip200 KO vascular tumour cells. a** Lysates from Ctrl, Fip200 KO tumour cells, and Fip200 KO tumour cells with OPN re-expression (Fip200 KO + OPN) were examined by western blots for various antibodies as indicated. Relative levels of pSTAT3/STAT3 and OPN (normalized to Ctrl cells) are shown as mean ± SD on the right. $n = 3$ independent experiments. **b**, **c** Representative images of colony formation assay of Ctrl, Fip200 KO and Fip200 KO + OPN tumour cells are shown in **b**, and mean ± SD of the colony number are shown in **c**. $n = 3$ biologically independent cells. **d**, **e** Cell migration of Ctrl, Fip200 KO and Fip200 KO + OPN tumour cells by wound healing assay. Representative images of cell migration at 0 and 16 h (**d**) and relative percentage of wound confluence (**e**) shown as mean ± SD. $n = 3$ biologically independent cells.

Scale bar in (**d**), 300 μm. **f**–**i** Ctrl, Fip200 KO and Fip200 KO + OPN tumour cells ($2 \times 10^6$ cells) were subcutaneously injected in recipient nude mice. Representative images of tumour harvested at 10 weeks after injection are shown in **f**. Mean ± SEM of tumour volume at 10 weeks after injection is shown in (**g**) ($n = 7$ mice for each group). Representative images of IHC for P-Stat3, Ki67, OPN and CD31 of tumour sections are shown in (**h**). Scale bar, 50 μm for OPN staining, 200 μm for others in (**h**). Quantification of P-Stat3 and Ki67 positive cells per field of view in Ctrl, Fip200 KO and Fip200 KO + OPN transplanted tumour sections are shown as mean ± SD in (**i**). $n = 4$ biologically independent samples. Unpaired two-tailed $t$ test was used in (**a**) (middle and right panels), (**c**, **e**, and **i**). One-way ANOVA with Tukey's multiple comparisons test was used in (**g**). For (**b**–**e**), at least two independent experiments were repeated. Source data are provided as a Source Data file.

Jak3-Forward: ACACCTCTGATCCCTCAGC;
Jak3-Reverse: GCGAATGATAAACAGGCAGGATG.
Lpar2-Forward: TGGTCACACTCATCGTGGGT;
Lpar2-Reverse: CCATGCGTGAGCAACTGTC.
Il15-Forward: ACATCCATCTCGTGCTACTTGT;
Il15-Reverse: GCCTCTGTTTTAGGGAGACCT.
Ptgs2-Forward: TTCAACACACTCTATCACTGGC;
Ptgs2-Reverse: AGAAGCGTTTGCGGTACTCAT.
β-actin-Forward: GGCTGTATTCCCCTCCATCG;
β-actin-Reverse: CCAGTTGGTAACAATGCCATGT.

## Western blotting

Lysates were prepared from tumor cells and analyzed by western blotting. Primary antibodies include FIP200 (1:1000, Cell Signaling #12436), p62 (1:2000, Cell Signaling #39749), TSC1 (1:1000, Cell Signaling #6935), phospho-ULK1 (Ser757) (1:1000, Cell Signaling #6888), ULK1 (1:1000, Cell Signaling #8054), LC3B (1:2000, Cell Signaling #43566), phospho-p70 S6K (Thr389) (1:1000, Cell Signaling #9205), p70 S6K (1:1000, Santa Cruz #sc-230), Atg5 (1:1000, Cell Signaling #12994), Atg7 (1:1000, Cell Signaling #8558), phospho-Stat3 Y705 (1:1000, Cell Signaling #9145), Stat3 (1:1000, Cell Signaling #3139), Osteopontin (1:1000, R&D Systems #MAB808), GAPDH (1:5000, Cell Signaling #5174), β-actin (1:10000, Sigma-Aldrich #), Vinculin (1:10000, Sigma-Aldrich #V4505) and HRP-linked anti-Rabbit IgG or anti-Mouse IgG (1:10000, Cell Signaling) were used as secondary antibodies. ECL HRP substrate (Thermo Scientific) was used for signal imaging.

## Cell immunofluorescence

A total of 20,000 cells were seeded into an 8-well chamber one day prior to treatment. Then, cells were fixed in 4% paraformaldehyde (PFA) for 10 min at room temperature (RT), followed with 1% BSA blocking buffer including 0.1% TritonX-100 for 30 min at RT. Phospho-Stat3 Y705 antibody was incubated with cells in the blocking buffer overnight at 4 °C. Next, Alexa fluor 594 donkey anti-rabbit IgG (1:1000, Jackson ImmunoResearch, #711-585-152) was used for 1 h at RT in dark, followed by ProLong® Gold Antifade Mountant with DAPI (Invitrogen #P36962) for nuclei staining and mounting. Images were acquired with Zeiss LSM confocal 710.

## Histology and immunostaining

Mouse tumor tissues were kept in 4% FPA at 4 °C. The specimens were then processed, embedded in paraffin and sectioned at 5μm. Heat-induced antigen retrieval was performed in citrate buffer using a pressure cooker. For hematoxylin and eosin (H&E) staining, sections were processed by standard H&E procedure followed with mount. For immunostaining, sections were incubated with primary antibodies at 4 °C overnight. Primary antibodies used were Ki67 (1:500, Spring Bioscience M3060), CD31 (1:500, Cell Signaling #77699), CD31 (1:50, Dianova DIA-310), phospho-S6RP (Ser240/244) (1:8000, Cell Signaling #5364), cleaved caspase-3 (1:100, Cell Signaling #9661), phospho-Stat3 Y705 (1:100, Cell Signaling #9145), Osteopontin (1:400, Kerafast #ENH094-FP). For immunohistochemistry staining, sections were then incubated with biotinylated goat anti-rabbit IgG (1:1000, Jackson ImmunoResearch, #111-065-003) for one hour at room temperature, followed with VECTASTAIN ABC peroxidase (Vector Laboratories) and DAB staining (Cell Signaling #8059), and counterstained with haematoxylin. For immunofluorescence staining, sections were incubated with Alexa Fluor 594 donkey anti-rabbit (1:1000, Jackson ImmunoResearch, #711-585-152) and Alexa Fluor 488 donkey anti-rat IgG (1:1000, Jackson ImmunoResearch #712-545-150) for one hour at RT in dark, and nuclei was counterstained with ProLong® Gold Antifade Mountant with DAPI. Images were acquired with Olympus BX41 Microscope. Six random fields from three different tissues in each group were quantified using ImageJ software.

## RNA-sequencing and bioinformatic analysis

Total RNA was isolated from cultured tumor cells using GeneJET RNA purification Kit (Thermo Scientific, REF K0731). Library preparation was conducted with NEBNext Ultra II Directional RNA Library Prep kit. Adapter trimmed reads in fastq format was used as our report. Demultiplexing is performed under Illumina BaseSpace default setting. Adapter: AGATCGGAAGAGCACACGTC; Adapter-Read2: AGATCGGAAGAGCGTCGTGT. Sequence reads were aligned to the reference genome using the TopHat2 aligner[65], and reads aligning to each known transcript were counted using Bioconductor packages for next-generation sequencing data analysis[66]. The differential expression analysis between different sample types was performed using the negative binomial statistical model of read counts as implemented in the *edgeR* Bioconductor package[67]. Differentially expressed genes were defined as genes with log2 fold change ≥1 or ≤ −1 with false discovery rate (FDR) cutoff 0.1. KEGG pathway analysis was performed using DAVID Bioinformatics Resources 6.8.

## Statistical analysis

Statistical significance was determined by unpaired two-tailed Student $t$ test, Bonferroni test, Log-rank (Mantel-Cox) test, or One-way ANOVA with Tukey's multiple comparisons test or Dunnett's multiple comparisons test. $P$ value < 0.05 was considered as statistical significance.

## Reporting summary

Further information on research design is available in the Nature Portfolio Reporting Summary linked to this article.

## Data availability

The RNA-sequencing data generated in this study have been deposited in the NCBI's Gene Expression Omnibus (GEO) under accession code GSE193568. All relevant data to evaluate the conclusions in the paper are within the paper and/or the Supplementary Information. Source data are provided with this paper.

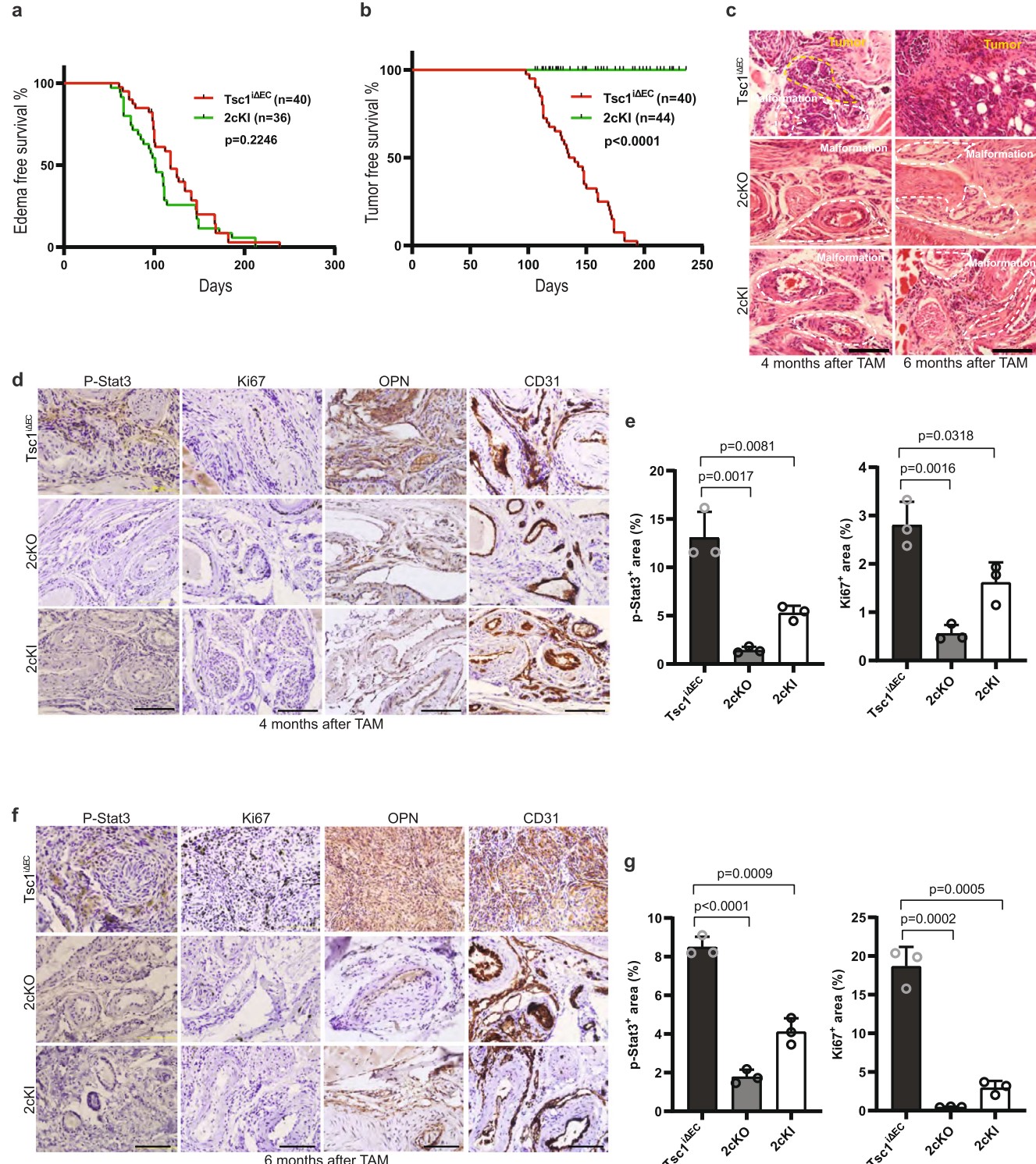

**Fig. 7 | Specific disruption of FIP200-mediated autophagy blocks LM progression to LAS. a** Kaplan-Meier analysis of edema development for *Tsc1*[iΔEC] (*n* = 40) and 2cKI (*n* = 36) mice. Log-rank (Mantel–Cox) test, *p* = 0.2246. **b** Kaplan-Meier analysis of LAS for *Tsc1*[iΔEC] (*n* = 40) and 2cKI (*n* = 44) mice. Log-rank (Mantel–Cox) test, *p* < 0.0001. **c** Representative H&E staining of vascular lesions of *Tsc1*[iΔEC], 2cKO and 2cKI mice at 4 and 6 months after TAM. Scale bar, 100 μm. **d, e** IHC for P-Stat3, Ki67, OPN and CD31 of vascular lesions in *Tsc1*[iΔEC], 2cKO and 2cKI mice at 4 months after TAM. Representative images (**d**) and quantification of

P-Stat3 and Ki67 positive cells per field of view in these mice (**e**) shown as mean ± SD. *n* = 3 biologically independent samples. Scale bar, 200 μm. **f, g** IHC for P-Stat3, Ki67, OPN and CD31 of vascular lesions in *Tsc1*[iΔEC], 2cKO and 2cKI mice at 6 months after TAM. Representative images (**f**) and quantification of P-Stat3 and Ki67 positive cells per field of view in these mice (**g**) shown as mean ± SD. *n* = 3 biologically independent samples. Scale bar, 200 μm. Log-rank (Mantel-Cox) test was used in (**a** and **b**). Unpaired two-tailed t test was used in (**e** and **g**). Source data are provided as a Source Data file.

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

## Acknowledgements

We would like to thank Mike Haas and Rose Copley for technical assistance, Dr. Mario Medvedovic, Jenny Chen (University of Cincinnati) and Xiaoting Zhu (Cincinnati Children's Hospital Medical Center) for bioinformatic analysis, Lindsey Huether and Glenn Doerman for graphics support and preparation of figures. We are grateful to Drs. Chenran Wang, Syn Yeo and members of the Guan lab for critical comments and suggestions in the preparation of this manuscript. This research is supported by NIH R01 HL073394, R01 CA211066 and R01 NS094144 to J.L.G.

## Author contributions

F.Y. and J.-L.G. designed the studies, F.Y. performed most of the studies, S.K., B.R. and Z.B. performed some studies, S.S. contributed reagents and assisted in the early stage of the study. F.Y., S.K. and B.R. collected and analyzed data, F.Y. and J.-L.G. wrote the manuscript text. All authors reviewed the manuscript.

## Competing interests

The authors declare no competing interests.
