## [Peer Review File · Nature Communications]

Autophagy inhibition prevents lymphatic malformation progression to lymphangiosarcoma by decreasing osteopontin and Stat3 signalingREVIEWER COMMENTS

Reviewer #1 (Remarks to the Author): with expertise in autophagy, cancer

In this manuscript, Yang and colleagues describe autophagy as a critical node in the development of malignant lymphangiosarcoma, primarily they argue through disruption of the OPN/Jak/Stat signalling pathway. The authors use a combination of genetic mouse models and cell line editing (shRNA and CRISPR/CAS9), together making a clear case for the requirement of autophagy for both progression of LM to LAS and its requirement for maintenance of tumour cell fitness.

The dKO phenotype in mice is convincing, supporting the critical role of autophagy in the LM-LAS transition. However, it is not clear how the autophagy activity is regulated in the Tsc1 null context, where autophagy is expected to be strongly inhibited. Are the authors suggesting that a residual autophagy activity plays the role, or the autophagy activity somehow escapes the mTOR-mediated inhibition during tumorigenesis? There is no data testing the autophagy activity/flux in tissues and even in culture. For example, in Fig. 1a, p62 doesn't appear to be high in Tsc1-ko compared to control (they need more controls). This could be due to 'bulk' tissues but still it would be informative if they provide other basic information about the autophagy pathway - p-Ulk1, LC3 etc.

Similarly, Fig. 1a should include basic autophagy characterization. In culture, they can easily measure the autophagy flux. (please define 'EC' in the first lane.)

The majority of in vitro and nude mice data is based on one spontaneously generated cell line. I would strongly encourage the authors to use additional cell lines/systems.

I wonder how autophagy blockade represses Opn transcription. The data are highly correlative, and it is not clear how direct it is. For example, it is possible that Opn low expressing cells may have an advantage in the absence of autophagy. Rescue experiments (e.g. Fip200 into Fip200-KO cells, and atg7 into atg7 KO cells etc) would be useful to address this question at least partially - if it fails to restore Opn, it is likely indirect, if it restores Opn, it might be more direct.

It would be important to show in all autophagy KO and KD cell lines alterations in p-Stat3.

Minor points:

No information on how survival was determined for the main cohorts. Were they culled due to edema? Or tumour? Or both? Was tumour free survival based on culling due to macroscopic tumour or due to a combination of that and histology in the edema culled mice. If histology what tissues and what tumours were found (I assume all were LAS and most in the liver)?

CRISPR KO and control cells: are they single clones?

Figure 1E: No description of tissue. Additionally, some supplemental images of different tissue (liver, cutaneous, lung(?)) would be of great help in understanding the model, including low and high mag images, with Cd31 staining. How did the liver look in the 2cKO mice, any further abnormalities? In particular, the original model describes a penetrant liver phenotype.

No information on how many tumour cells were injected (e.g. fig. 2f)?

Figure 2 H and I would benefit from higher magnification images side by side with the low mag images.

Fig 3D: It is not clear if both of the Fip200 KO lines were used in injections?

Figure 4: D-E should also contain information from Atg5 and Atg7 KO.

Fig 4C: How much overlap did Atg5 and atg7 (or other intersects) have?

Reviewer #2 (Remarks to the Author): with expertise in autophagy, cancer

In this manuscript, Yang et al conditionally deleted essential autophagy gene Fip200 in Tsc1 Δ EC mouse model for human LAS and found that deletion of FIP200 or specifically disrupting its autophagy function inhibited the development of malignant lymphangiosarcoma (LAS) from the benign Lymphatic malformation (LM) lesions. This study suggests that inhibiting autophagy could potentially prevent tumorigenesis for this deadly disease. They further identified autophagy-dependent expression of osteopontin (OPN) and its autocrine stimulation of Jak/Stat3 signaling as a key mediator in the promotion of vascular tumor cell proliferation and tumorigenicity. So far, LM is a risk factor for LAS and the effective treatment for the deadly LAS is urgently needed. The findings from this study suggest that targeting autophagy and/or its downstream OPN expression will have important clinical implications for LM patients. It's commonly believed that autophagy prevents tumor initiation. The data from this study suggest that autophagy can promote LAS initiation, which is important for our deeper understanding of the role of autophagy in cancer.

Please provide these revisions before considering publication in Nature Communication:

1. For cell migration assay, instead of providing quantified bar graph, please also provide migration image of scratch assay.
2. In Both GEMM and xenograft models (Fig. 1 &2), Fip200 ablation inhibits cell proliferation. Does Fip200 deletion or knockdown increase more cell death, such as apoptosis?
3. Colony formation was provided to assess the cell proliferation when Fip200 was knockout or knockdown. Please also provide cell proliferation assay in culture medium.
4. For Western blot results, please also show the quantification after normalizing with loading control in Fig 5E and 6A.
5. Fip200 KO impairs LAS tumor growth, but had no effect on mTOR activity (Fig. 2K). it will be interesting to know the mTOR activity in OPN KO allograft tumors (Fig. 4).
6. For IHC in Fig. 6 and Fig. 7, please also show the quantification of p-Stat3. Please provide Ki67 IHC quantification of Fig 1F.

Reviewer #3 (Remarks to the Author): with expertise in sarcoma, models

The study by Tang and colleagues uses a mouse model driven by deletion of Tsc1 to understand the drivers of malignant progression of lymphatic malformation to lymphangiosarcoma. They provide convincing evidence in support of a pro-tumorigenic role for FIP200 in lymphangiosarcoma development that its linked to its role in autophagy regulation.

The following points need to be addressed:

1. Figure 1 A describes the deletion of FIP200 in lung endothelial cells. What is the efficiency of deletion in the model? There is still significant expression of FIP200. Have they looked at FIP200 expression in the extended vessels of the 2cKO mice?
2. Does FIP200 expression in the endothelial cells change as tumors develop in the Tsc1 KO mice?
3. Figure 1E – details of where these tumors developed should be included in the figure legend
4. Details of the controls used for the shRNA knockdown of Fip200 is missing.
5. The authors should provide a list of the genes that are common to the Fip200, Atg5 and Atg7 KO cell lines with details of the significance values. Are the genes shown in Figure 4D common to all 3 lines? Why were these chosen for validation?
6. It is not clear what the image in Figure 4F is showing. Which cells are expressing OPN?
7. What are the levels of pSTAT3 in the ATG7 and ATG5 KO cells? Is the involvement of Stat3 Fip200 specific or can they support the claim that the pathway is involved in regulation of tumorigenicity by autophagy? A western blot showing relative expression of pSTAT3 in all the cell lines (Fip200, ATG5, ATG7, OPN KO) should be included along with quantification from biological replicates of the changes in phosphorylation of STAT3 and levels of OPN. What phosphorylation site in STAT3 has been measured? Is there evidence that the STAT3 pathway is activated ie translocation to the nucleus/induction of STAT3 dependent gene expression?
8. The link between Fip200, OPN and STAT3 phosphorylation is correlative. It is likely that a number of different signaling pathways are altered (as indicated by the RNASeq data). Is blockade

of STAT3 sufficient to ablate the tumorigenicity of the 562 cells? Does inhibition of STAT3 prevent the rescue of the in vitro phenotypes in the OPN re-expressed cells? The authors cannot claim that they have "identified autophagy-dependent expression of OPN and its autocrine stimulation of Jak/Stat3 signaling as a key mediator in the promotion vascular tumor cell proliferation" based on the data shown.

9. What is the relationship between STAT3 signaling, OPN and Fip200 in human patient samples?

10. Figure 6H: it is not clear what the images are representing. From the tumor volume measurements there appears to be very little tumor growth following injection of the Fip200 KO cells so in the image shown does this correspond to tumor? How accurate is the Ki67 quantification? Is it only analysis of tumor cells? There appears to be large Ki67 areas in the Fip200 tumors.

11. The authors have used tumorigenicity of the 562 malignant cells when injected subcutaneously into mice as a surrogate for malignant progression of lymphatic malformation to lymphangiosarcoma. Can they show activation of STAT3 and levels of OPN in the spontaneous Tsc1f/f mouse model in support of its importance in driving malignant conversion?

Reviewers' Comments:

Reviewer #1:

1. The dKO phenotype in mice is convincing, supporting the critical role of autophagy in the LM-LAS transition. However, it is not clear how the autophagy activity is regulated in the *Tsc1* null context, where autophagy is expected to be strongly inhibited. Are the authors suggesting that a residual autophagy activity plays the role, or the autophagy activity somehow escapes the mTOR-mediated inhibition during tumorigenesis? There is no data testing the autophagy activity/flux in tissues and even in culture. For example, in Fig. 1a, p62 doesn't appear to be high in *Tsc1*-ko compared to control (they need more controls). This could be due to 'bulk' tissues but still it would be informative if they provide other basic information about the autophagy pathway - p-Ulk1, LC3 etc.

New data are included to show Western blotting for *Tsc1*, P-ULK1 at S757, and LC3B from lung endothelial cell (EC) samples of more mice (4 each for wild type as Ctrl, *Tsc1*^{iAEC} and 2cKO mice (new Fig. 1A), as suggested.

Although it is well known that mTORC1 activation negatively regulates autophagy in many cells, recent studies suggested that some cancer and other cells maintain autophagy activity which are required to sustain hyperactivation of mTORC1 caused by *Tsc*-deficiency (e.g. Wang et al. Nat Metab 1, pp1127-1140, 2019; Wang et al. British J of Cancer 124, pp1711-1723, 2021). Our data are consistent with the idea that autophagy activity somehow escapes the mTOR-mediated inhibition in *Tsc1*-deficient ECs. No increased p62 was observed in lung ECs from *Tsc1*^{iAEC} mice compared to those from Ctrl mice, suggesting that autophagy was not significantly inhibited. We do not think that the lack of change in p62 levels was due to "bulk" tissues in the sample (i.e. non ECs thus no *Tsc1* deletion by EC-specific Cre) because *Tsc1* blots confirmed deletion of *Tsc1* in the samples from *Tsc1*^{iAEC} and 2cKO mice (compared to those from Ctrl mice). Moreover, increased p62 were observed in lung ECs from 2cKO mice, indicating inhibition of autophagy in ECs of these mice. Lastly, we did not find significant phosphorylation of ULK1 at S757 (mTORC1 phosphorylation site to inhibit autophagy) in ECs from *Tsc1*^{iAEC} and 2cKO mice, which might explain why autophagy in ECs was not inhibited by *Tsc1* deletion in these cells.

However, the changes in LC3B levels were not significant, so we assessed autophagy flux in these cells through in vitro experiments as suggested by the reviewer, in point 2 below.

2. Similarly, Fig. 1a should include basic autophagy characterization. In culture, they can easily measure the autophagy flux. (please define 'EC' in the first lane.)

New data are included to show autophagy flux of lung endothelial cells from wild type, *Tsc1*^{iAEC} and 2cKO mice (pooled from 4 mice for each)(new Fig. S1), as suggested. (EC stands for endothelial cells, and the legend is revised accordingly) These results are consistent with findings in new Fig. 1A that autophagy is maintained despite *Tsc1* deletion and mTORC1 activation in ECs from both *Tsc1*^{iAEC} and 2cKO mice. Importantly, the autophagy flux under HBSS conditions in *Tsc1*^{iAEC} (differential between lanes 5 and 6) was diminished upon ablation of

FIP200 in 2cKO cells (differential between lanes 8 and 9). This indicates that ablation of FIP200 impairs autophagy flux in *Tsc1* deleted ECs

3. The majority of in vitro and nude mice data is based on one spontaneously generated cell line. I would strongly encourage the authors to use additional cell lines/systems.

We appreciate the comment and have used another immortalized tumor cell line (designated 5864 tumor cells) from tumors in *Tsc1*^{i^ΔEC} mice. New data are included to show that similar to 562 cells, ablation of FIP200 in 5864 cells also decreased cell proliferation, colony formation and cell migration in vitro, and abolished their tumorigenicity in nude mice (new Fig. S2), as suggested.

4. I wonder how autophagy blockade represses *Opn* transcription. The data are highly correlative, and it is not clear how direct it is. For example, it is possible that *Opn* low expressing cells may have an advantage in the absence of autophagy. Rescue experiments (e.g. *Fip200* into *Fip200*-KO cells, and *atg7* into *atg7* KO cells etc) would be useful to address this question at least partially - if it fails to restore *Opn*, it is likely indirect, if it restores *Opn*, it might be more direct.

New data are included to show rescue of *Opn* in *Fip200* KO cells upon ectopic expression of FIP200 (new Fig. 4E), suggesting that it might be more direct for autophagy regulation of *Opn* expression.

5. It would be important to show in all autophagy KO and KD cell lines alterations in p-Stat3.

New data are included to show decreased p-Stat3 upon autophagy blockade by depletion of FIP200, *Atg5* or *Atg7* (new Figs. 4I and 5F), as suggested. Please note that decreased p-Stat3 in FIP200 KO tumor cells was shown in previous Fig. 6A.

Minor points:

No information on how survival was determined for the main cohorts. Were they culled due to edema? Or tumour? Or both? Was tumour free survival based on culling due to macroscopic tumour or due to a combination of that and histology in the edema culled mice. If histology what tissues and what tumours were found (I assume all were LAS and most in the liver)?

For Fig. 1B, edema free survival was determined based on signs of edema appearance with mice paws and tails swelling. For Fig. 1D, tumor free survival was determined based on formation of solid cutaneous tumors in mice paws and/or tails. Liver malformations were mostly benign based on histological examination (Sun et al., *Cancer Cell* 28, p758-772) and were not taken into account for tumor free survival curves.

CRISPR KO and control cells: are they single clones?

Two clones of *Fip200* KO cells were analyzed initially with similar results (see Fig. 3A-3D), and one clone (*Fip200* KO-2) was used in subsequent studies. *Atg5* KO and *Atg7* KO are single clones. Two *OPN* KO clones were used for in vitro experiments, but one was used for transplant

experiments. Ctrl cells (i.e. 562 cells) is derived from the tumors developed in *Tsc1^{iΔEC}* mice as described in the paper.

Figure 1E: No description of tissue. Additionally, some supplemental images of different tissue (liver, cutaneous, lung(?)) would be of great help in understanding the model, including low and high mag images, with Cd31 staining. How did the liver look in the 2cKO mice, any further abnormalities? In particular, the original model describes a penetrant liver phenotype.

Fig. 1E shows vascular lesions in the paws of different mice. The figure legend is updated accordingly.

In the current study, we focused on the progression of lymphatic malformation to malignant lymphatic angiosarcoma in the paws and tails of *Tsc1^{iΔEC}* mice. Liver vascular tumours in *Tsc1^{iΔEC}* mice are benign or low-grade vascular tumors which are derived from blood vessel endothelial cells. Lung tissues in *Tsc1^{iΔEC}* mice had normal histological morphology shown in the previous study. Due to the benign nature of these malformations, liver and lung tissues were not examined in 2cKO mice in the current study, but it will be interesting to study potential changes of other tissues/organs upon FIP200 deletion in *Tsc1^{iΔEC}* mice in future studies.

No information on how many tumour cells were injected (e.g. fig. 2f)?

This information is added to the legends for these panels (new Figs. 2G [previous fig. 2f], 3E, 6F-6H and S3A), as suggested.

Figure 2 H and I would benefit from higher magnification images side by side with the low mag images.

New panels are included to show higher magnification images below the low mag images for Figs. 2H and 2I (new Figs. 2I and 2J in the revised paper), as suggested.

Fig 3D: It is not clear if both of the Fip200 KO lines were used in injections?

Only one clone is used in the injection. Please see more detailed explanation in minor point #2 above.

Figure 4: D-E should also contain information from Atg5 and Atg7 KO.

New data are included to show changes in the expression of multiple genes including Spp1 (new Fig. 4H) and decreased levels of Opn (new Fig. 4I) in Atg5 KO and Atg7 KO tumor cells, as suggested.

Fig 4C: How much overlap did Atg5 and atg7 (or other intersects) have?

This panel is updated to indicate the number of common genes in two out of the 3 KO cells (highlighted in green, new Fig. 4C), as suggested. Please note that we also corrected previous mistakes for the number of genes that are unique to individual KO cells (i.e. total of 27 genes,

not 28 genes, were decreased in Fip200 KO cells; more importantly, the unique genes should be 4 not 27 because the 27 genes included those overlapping with both Atg5 KO and Atg7 KO [20 genes highlighted by red], those overlapping with Atg5 KO but not Atg7 KO [1 in green], those overlapping with Atg7 KO but not Atg5 KO [2 in green], and those that are unique to Fip200 KO [4 in black]).

Reviewer #2:

1. For cell migration assay, instead of providing quantified bar graph, please also provide migration image of scratch assay.

New data are included to show migration image of scratch assay (new Fig. 2E), as suggested.

2. In Both GEMM and xenograft models (Fig. 1 &2), Fip200 ablation inhibits cell proliferation. Does Fip200 deletion or knockdown increase more cell death, such as apoptosis?

New data are included to show increased apoptosis upon FIP200 depletion in both GEMM (new Figs. 1F and 1G) and xenograft (new Figs. 2J and 2L) models, as suggested.

3. Colony formation was provided to assess the cell proliferation when Fip200 was knockout or knockdown. Please also provide cell proliferation assay in culture medium.

New data are included to show decreased cell proliferation after Fip200 knockout (new Fig. 3B), as suggested. Similar data following Fip200 knockdown was shown in Fig. 2B of the original manuscript.

4. For Western blot results, please also show the quantification after normalizing with loading control in Fig 5E and 6A.

New data are included to show quantification for Figs. 5E and 6A, as suggested.

5. Fip200 KO impairs LAS tumor growth, but had no effect on mTOR activity (Fig. 2K). it will be interesting to know the mTOR activity in OPN KO allograft tumors (Fig. 4).

New data are included to show the mTOR activity in OPN KO allograft tumors (new Fig. S4), as suggested. It is interesting to note decreased mTORC1 signaling in OPN KO tumor cells. Although mTORC1 signaling was not reduced after depletion of FIP200 (see Figs. 2J and 2M), future studies will be needed to clarify the potential contribution of mTORC1 signaling in the decreased tumorigenesis of vascular tumor cells upon autophagy blockade and reduced OPN expression. Discussion section is revised to include these as well as new data in support of a role for Stat3 signaling in mediating the regulation of vascular tumor cells by autophagy-dependent expression of OPN (p14 lines 5-11 and lines 13-19).

6. For IHC in Fig. 6 and Fig. 7, please also show the quantification of p-Stat3. Please provide Ki67 IHC quantification of Fig 1F.

New data are included to show quantification for p-Stat3 IHC in Figs. 6 and 7 (see new left side graphs of Figs. 6J, 7E and 7G), as suggested.

Please see Fig. 7G for quantification of Ki67 IHC of *Tsc1^{iΔEC}* and 2cKO mice (i.e. representative image in Fig. 1F).

Reviewer #3:

1. Figure 1 A describes the deletion of FIP200 in lung endothelial cells. What is the efficiency of deletion in the model? There is still significant expression of FIP200. Have they looked at FIP200 expression in the extended vessels of the 2cKO mice?

We thank the reviewer for the comment and prepared lung ECs from more mice for Western blot analysis for FIP200 and other proteins (new Fig. 1A). As in the previous samples, there are still expression of FIP200 in the samples of 2cKO mice, suggesting that FIP200 was not completely deleted in these cells. Alternatively, there may be some contamination of other cells (i.e. not ECs) in our samples which retained FIP200 expression. Nevertheless, EC-specific induction of Cre should mediate deletion of floxed FIP200 allele in the same cells as floxed *Tsc1* allele, which is responsible for driving the development of lymphatic malformation and progression to lymphangiosarcoma as described previously (Sun et al Cancer Cell 28, pp758-772). Indeed we observed reduced levels of FIP200 as well as increased levels of p62 (indicative of FIP200 deletion based on previous studies in other cells), consistent with at least partial deletion of FIP200 in these cells. Lastly the apparent phenotype of 2cKO mice (vs *Tsc1^{iΔEC}* mice) is also consistent with the idea that the extent of FIP200 deletion is sufficient even if not complete to induce the changes observed.

We are unable to examine FIP200 expression in the extended vessels of 2cKO mice, as there is currently no available antibody for FIP200 for immunostaining of tissue samples (despite repeated efforts from my lab testing various commercial antibodies for FIP200).

2. Does FIP200 expression in the endothelial cells change as tumors develop in the *Tsc1* KO mice?

As mentioned above, no suitable antibody for FIP200 is available for staining tissue sections (e.g. for tumors at 4 months vs 6 months as shown in Fig. 1E). Moreover, there is very little tumor cells at earlier stage (e.g. 4 months in Fig. 1E) to obtain sufficient protein extracts specifically from tumor cells for comparison by western blotting. To overcome these technical challenges, we prepared lysates from lung endothelial cells (i.e. before tumor development) and vascular tumors of *Tsc1^{iΔEC}* mice for comparison of FIP200 levels by Western blotting. We found higher levels of FIP200 in vascular tumors than in lung ECs, suggesting increased FIP200 expression as tumors develop in these mice. New results are included as Fig. A for the reviewer, and we will be happy to include it as a supplemental figure in the revised paper if the editor suggests so.

3. Figure 1E – details of where these tumors developed should be included in the figure legend

These tumors are in mouse paws, and the figure legend is revised accordingly.

4. Details of the controls used for the shRNA knockdown of Fip200 is missing.

The control used for Fip200 knockdown is a scrambled shRNA purchased from Addgene (plasmid #1864). This information is added in the Method section accordingly.

5. The authors should provide a list of the genes that are common to the Fip200, Atg5 and Atg7 KO cell lines with details of the significance values. Are the genes shown in Figure 4D common to all 3 lines? Why were these chosen for validation?

New data are included to show a list of the genes in PI3K-Akt signaling, ECM-receptor interaction, and TNF signaling that are common to the Fip200, Atg5 and Atg7 KO cell lines (i.e. reduced when compared to Ctrl) with details of the significance values (new Supplemental Table S1), as suggested.

The genes in Fig. 4D are common to all 3 lines. They were selected as representatives of the down-regulated genes in FIP200 KO cells (i.e. similar to *Spp1* which is the focus of the current paper) for validation of results by RNA-seq.

6. It is not clear what the image in Figure 4F is showing. Which cells are expressing OPN?

Fig. 4F shows immunohistochemical staining of tumor sections for OPN, as indicated in the legend. New data are included to show double label immunofluorescent staining of the sections for CD31 and OPN, confirming that vascular tumor cells are expressing OPN (new Fig. 4G), as suggested.

7. What are the levels of pSTAT3 in the ATG7 and ATG5 KO cells? Is the involvement of Stat3 Fip200 specific or can they support the claim that the pathway is involved in regulation of tumorigenicity by autophagy? A western blot showing relative expression of pSTAT3 in all the cell lines (Fip200, ATG5, ATG7, OPN KO) should be included along with quantification from biological replicates of the changes in phosphorylation of STAT3 and levels of OPN. What phosphorylation site in STAT3 has been measured? Is there evidence that the STAT3 pathway is activated ie translocation to the nucleus/induction of STAT3 dependent gene expression?

New data are included to show decreased p-Stat3 upon autophagy blockade by depletion of FIP200, Atg5 or Atg7 (new Figs. 4I and 5F), as suggested. Please note that decreased p-Stat3 in OPN KO and FIP200 KO tumor cells were shown in previous Fig. 5E and Fig. 6A, respectively.

Fig. A. Lysates were prepared from lung endothelial cells and vascular tumors of three different *Tsc1*^{ΔEC} mice and analyzed by immunoblotting with various antibodies as indicated.

The antibody is for phosphorylated Y705 of mouse Stat3 and this information is updated in the method section of the revised paper.

New data are also included to show p-Stat3 translocation to the nucleus (indicative of the activation of STAT3 pathway) was decreased upon autophagy blockade by depletion of FIP200, Atg5 or Atg7, and this decrease was reversed by Opn addition to Fip200 KO tumor cells (new Fig. 5G), as suggested.

8. The link between Fip200, OPN and STAT3 phosphorylation is correlative. It is likely that a number of different signaling pathways are altered (as indicated by the RNASeq data). Is blockade of STAT3 sufficient to ablate the tumorigenicity of the 562 cells? Does inhibition of STAT3 prevent the rescue of the in vitro phenotypes in the OPN re-expressed cells? The authors cannot claim that they have “identified autophagy-dependent expression of OPN and its autocrine stimulation of Jak/Stat3 signaling as a key mediator in the promotion vascular tumor cell proliferation” based on the data shown.

We agree with the reviewer and have revised this statement to “Our data also suggested that autophagy-dependent expression of OPN and its autocrine stimulation of Jak/Stat3 signaling contributed to the promotion of vascular tumor cell proliferation” (see the first paragraph of “Discussion” section (p13 lines 9-11)).

9. What is the relationship between STAT3 signaling, OPN and Fip200 in human patient samples?

We thank the reviewer for this question. Unfortunately, we do not have human patient samples to perform the studies currently. In our previous studies published in Cancer Cell in 2015, Sun et al used human samples showing relationships of several other signaling pathways. However, the limited number of tissue microarrays from a previous collaborator were used up and no longer available to us now. In addition, there is currently no available antibody for FIP200 for immunostaining of tissue samples (despite repeated efforts from my lab testing various commercial antibodies for FIP200). We will be happy to include discussion on this point as a potential caveat of the current study, if the reviewer and/or editor suggest us to do so. We will perform this interesting study when suitable samples and reagents become available in the future.

10. Figure 6H: it is not clear what the images are representing. From the tumor volume measurements there appears to be very little tumor growth following injection of the Fip200 KO cells so in the image shown does this correspond to tumor? How accurate is the Ki67 quantification? Is it only analysis of tumor cells? There appears to be large Ki67 areas in the Fip200 tumors.

We believe the reviewer refers to Fig. 6I in the original submission. The Fip200 KO samples shown in the image are from tumor developed in these mice. Although FIP200 deletion significantly inhibited tumor cell growth, we could detect smaller tumors in a fraction of the recipient mice. Panels 6F and 6G show one representative experiment. Panel H include more samples (and one larger than the tumor shown in Fig. 6G).

We believe staining are from tumor cells, as they are also positive by CD31 staining, although we could not completely exclude the possibility that some tumor stroma cells are present in the sample and/or show Ki67+ staining.

Although there are some Ki67 positive area in Fip200 KO tumors, they are smaller than Ctrl or Fip200 KO + OPN tumors as shown in Fig. 6J. The panel in Fig. 6I is replaced by one that is a better representative image.

11. The authors have used tumorigenicity of the 562 malignant cells when injected subcutaneously into mice as a surrogate for malignant progression of lymphatic malformation to lymphangiosarcoma. Can they show activation of STAT3 and levels of OPN in the spontaneous *Tsc1f/f* mouse model in support of its importance in driving malignant conversion?

Figs. 7D and 7F show activation of STAT3 (P-Stat3 staining) in the spontaneous *Tsc1f/f* mouse model (i.e. *Tsc1^{iΔEC}* mice), supporting its importance in driving malignant conversion. We also added new data showing OPN levels providing further support (updated Figs. 7D and 7F), as suggested.

REVIEWER COMMENTS

Reviewer #1 (Remarks to the Author):

Although some key experiments still rely on the single cell line (e.g. opn-ko, Fig. 5), the authors have addressed most of the reviewers' concerns with substantial new experimental data.

Reviewer #2 (Remarks to the Author):

All my previous concerns have been addressed in this revised manuscript. I don't have any other concern for the publication in Nature Communications.

Reviewer #3 (Remarks to the Author):

The authors have included additional samples in the western blot in Fig 1a showing reduced expression of FIP200 in endothelial cells from the 2cKO mice which they describe in the text. There also appears to be reduced expression in the Tsc1 deleted endothelial cells. Is this correct? Quantification of FIP200 expression relative to the loading control should be included for Fig 1 and discussed in the text.

I am also struggling to see any difference between the LC3I/II levels in the 2cKO and Tsc1 deleted endothelial cells in Supplementary Figure 1 to support their claim that FIP200 is altering autophagy flux in the Tsc1 deleted endothelial cells. Again quantification should be carried out and this point needs to be addressed if there are no differences.

The authors have addressed all the other points raised.

Reviewer #1 (Remarks to the Author):

Although some key experiments still rely on the single cell line (e.g. opn-ko, Fig. 5), the authors have addressed most of the reviewers' concerns with substantial new experimental data.

We thank the reviewer for his/her comments and have used another immortalized tumor cell line (designated 5864 tumor cells) for additional studies in previous Fig. 5. New data are included to show that similar to 562 cells, ablation of OPN in 5864 cells also decreased cell proliferation, colony formation in vitro, tumorigenicity in nude mice, and Stat3 phosphorylation (new Fig. S4), as suggested.

Reviewer #2 (Remarks to the Author):

All my previous concerns have been addressed in this revised manuscript. I don't have any other concern for the publication in Nature Communications.

We thank the reviewer for her/his support of the revised manuscript for publication in Nature Communications.

Reviewer #3 (Remarks to the Author):

The authors have included additional samples in the western blot in Fig 1a showing reduced expression of FIP200 in endothelial cells from the 2cKO mice which they describe in the text. There also appears to be reduced expression in the Tsc1 deleted endothelial cells. Is this correct? Quantification of FIP200 expression relative to the loading control should be included for Fig 1 and discussed in the text.

New data are included to show quantification of FIP200 expression relative to the loading control (new Fig. 1A right panel), as suggested. We also replaced previous Fig. 1A (new Fig. 1A left panel) with representative image from another experiment.

I am also struggling to see any difference between the LC3I/II levels in the 2cKO and Tsc1 deleted endothelial cells in Supplementary Figure 1 to support their claim that FIP200 is altering autophagy flux in the Tsc1 deleted endothelial cells. Again quantification should be carried out and this point needs to be addressed if there are no differences.

We thank the reviewer for the suggestion. We performed additional experiments and replaced previous Fig. S1 with representative image from another experiment (new Fig. S1A). We also quantified the results and included new data to show decreased autophagy flux upon FIP200 deletion in Tsc1-null ECs (new Fig. S1B), as suggested.

The authors have addressed all the other points raised.

Thank you.

REVIEWERS' COMMENTS

Reviewer #1 (Remarks to the Author):

The authors have adequately addressed the reviewers' questions.

One minor point: they have added quantitative data on autophagy flux (Fig. S1B), but I don't find the method for the quantification of autophagy flux. It would be helpful for the audience to include it in the method section.

Reviewer #1 (Remarks to the Author):

The authors have adequately addressed the reviewers' questions.

One minor point: they have added quantitative data on autophagy flux (Fig. S1B), but I don't find the method for the quantification of autophagy flux. It would be helpful for the audience to include it in the method section.

We would like to thank the reviewer's comment noting that we have adequately addressed the reviewers' questions. We also appreciate one minor point by the reviewer and now have added the method for the quantification of autophagy flux for Fig. S1B in the method section (see p19, near bottom tracked version; p18 lines 12-15, clean version), as suggested.